# KOALA: Koopman Operator Learning for WiFi-Based Anticipatory Human Motion Prediction

## Abstract

WiFi Channel State Information (CSI) has emerged as a privacy-preserving alternative to cameras for human pose estimation. However, existing approaches treat pose inference as an instantaneous regression problem and do not model temporal dynamics, making future motion prediction infeasible. Naively applying vision-based prediction methods compounds the estimation noise already present in CSI-derived poses, as autoregressive rollouts amplify errors at every step. We propose KOALA, the framework for human motion prediction directly from WiFi CSI, by lifting noisy CSI-derived pose sequences into a learned Koopman latent space where nonlinear dynamics become linear, enabling multi-horizon prediction via simple matrix-vector products without autoregressive iteration or error accumulation. A residual CSI-conditioned operator resolves the identity attractor problem inherent from Koopman formulations, and an anchor-delta prediction head eliminates the degenerate shortcut of copying the current pose across all horizons. To regularise the lifting and operator jointly, we introduce a Koopman Anchored Latent (KAL) loss that operates in the temporal-encoder feature space, enforcing dynamical consistency across prediction horizons without requiring contrastive, spectral, or auxiliary losses. Experiments on MM-Fi and WiPose show that KOALA achieves robust, consistent performance across both short- and long-term prediction horizons, outperforming all baselines by a substantial margin.

FIX

## 1 Introduction

Anticipating human motion before it fully unfolds is a fundamental capability for intelligent systems operating in human-centric environments, enabling proactive assistance in smart homes, collision avoidance in human-robot collaboration, and early intervention in healthcare monitoring (Lyu et al., 2022). The dominant paradigm for this task relies on skeletal sequences derived from optical motion capture or RGB-D sensors (Martinez et al., 2017; Mao et al., 2019; 2020; Xu et al., 2023; Guo et al., 2023). Despite remarkable progress, camera-based systems impose significant deployment constraints: they require line-of-sight, degrade under occlusion and low illumination, and raise pervasive privacy concerns in domestic and clinical settings. These

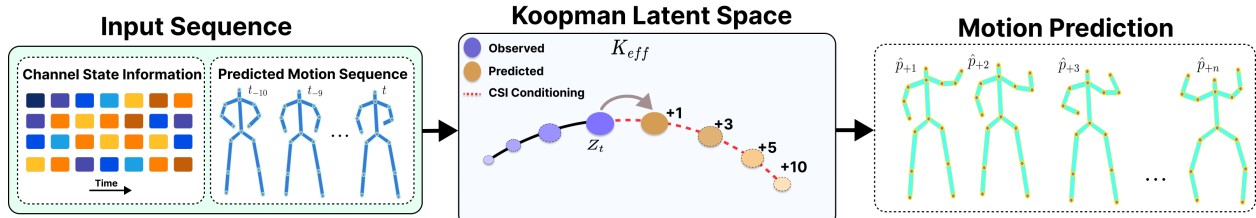

Figure 1: KOALA predicts future human motion from WiFi CSI and observed skeleton poses, these are lifted into a Koopman latent space, where a single CSI-conditioned linear operator propagates the state to multiple future horizons without autoregressive iteration.

barriers motivate a fundamentally different sensing modality, which is ubiquitous, infrastructure-free, and inherently privacy-preserving.

WiFi Channel State Information (CSI) has emerged as a compelling alternative for non-intrusive human sensing (Zhou et al., 2023). By capturing how the human body perturbs multipath radio-frequency propagation, CSI encodes rich motion signatures without requiring any wearable device or visual access to the monitored space (Zhou et al., 2022; Wang et al., 2015; Yang et al., 2023a). A growing line of work has demonstrated the viability of recovering instantaneous 3D skeletal poses from CSI signals, leveraging cross-modal supervision from synchronized cameras to bridge the modality gap (Chen et al., 2023; Yang et al., 2022). Subsequent advances have pushed toward 2D and 3D reconstruction (Zhou et al., 2022; 2023), multi-person scenarios (Yan et al., 2024), and cross-environment generalization (Zhou et al., 2024). These systems, however, share a critical architectural assumption, they operate as instantaneous regressors, mapping a fixed CSI observation window to the current pose state. Modeling forward temporal dynamics, predicting where a person will be rather than where they are, remains largely unexplored. Despite its promise, forecasting human motion from WiFi CSI is fundamentally more challenging than its vision-based counterpart. First, CSI-derived pose observations are inherently noisy and incomplete due to multipath fading, environmental interference, and hardware variability, leading to spatial jitter and missing joints. Second, unlike clean motion capture sequences, the temporal dynamics extracted from CSI are less stable and harder to model. Third, multi-horizon prediction requires maintaining temporal consistency over long sequences, where autoregressive strategies suffer from error accumulation and exposure bias. These challenges call for new modeling paradigms that are robust to noise while capturing long-term motion dependencies.

To address these gaps, we propose **K**oopman **O**perator with **A**ttentive **L**ifting **A**rchitecture (**KOALA**), the first unified framework for WiFi-CSI-based human motion prediction (see Fig. 1). Moving beyond conventional approaches that focus on instantaneous pose estimation or action recognition (Huang et al., 2024; Zhou et al., 2024), KOALA forecasts future human motion directly from ambient Radio Frequency (RF) signals by lifting noisy CSI-derived pose sequences into a learned Koopman latent space where nonlinear dynamics become linear. Multi-horizon prediction is then performed via simple matrix-vector products, without autoregressive iteration or error accumulation. To enforce dynamical consistency and representational richness, KOALA is trained with a Koopman Anchored Latent (KAL) loss that operates in the temporal-encoder feature space, shaping the lifting by minimizing prediction errors at the timescales relevant to the task. Our contributions are summarized as follows.

- We introduce a novel framework for multi-horizon human pose prediction directly from WiFi CSI, establishing a new problem at the intersection of RF sensing and anticipatory motion understanding.

- We propose a residual CSI-conditioned Koopman operator that resolves the identity attractor problem inherent in prior Koopman-based methods, paired with an anchor-delta prediction formulation that eliminates the degenerate shortcut of copying the current pose across all horizons.

- We introduce a Koopman Anchored Latent loss that subsumes reconstruction, linearity, and stability objectives into a single anchored feature-space prediction term, removing the need for separate contrastive, spectral, and auxiliary losses used in prior work.

- We conduct extensive experiments on MM-Fi (Yang et al., 2023b) and WiPose (Zhou et al., 2022), demonstrating that KOALA consistently outperforms existing baselines across both short- and long-term prediction horizons. Code will be released upon publication.

## 2 Related Work

### 2.1 WiFi CSI-Based Human Pose Estimation

Early works established cross-modal supervision as the dominant paradigm for WiFi-based human pose estimation, using synchronized cameras as teacher signals to map CSI amplitude and phase to 2D joint coordinates (Chen et al., 2023; Yang et al., 2022). Subsequent research extended this to full 3D reconstruction, WiPose (Jiang et al., 2020) combines residual CNNs with recurrent models for spatiotemporal feature

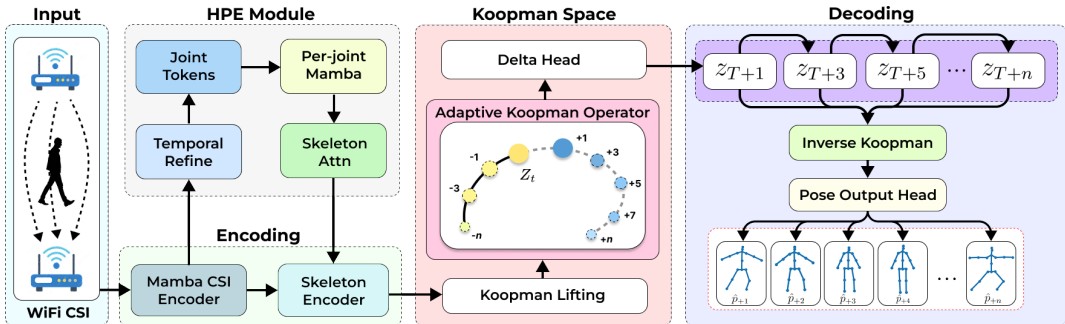

Figure 2: Overview of the KOALA framework. WiFi CSI is encoded by a Mamba CSI encoder whose features are passed to the HPE module and skeleton encoder. The HPE module estimates poses via temporal refinement, joint token construction, per-joint Mamba, and skeleton-biased attention; estimated poses are gradient-detached before entering the skeleton encoder. Dual-stream fusion and a temporal encoder generate contextual motion features, which are lifted into the Koopman latent space. A residual CSI-conditioned operator propagates latent states across prediction horizons without autoregression. Inverse lifting and a delta head decode pose deltas, which are added to an anchor pose for final predictions.

extraction, while MetaFi++ (Zhou et al., 2023) employs Transformer encoders for richer motion representations. Person-in-WiFi 3D (Yan et al., 2024) further scales to multi-person scenarios via set prediction, and AdaPose (Zhou et al., 2024) addresses cross-environment generalization through domain-invariant adaptation. Despite these advances, all existing systems work as instantaneous regressors, mapping a fixed CSI window to the current pose without modeling forward temporal dynamics. Predicting future skeletal states from WiFi sequences remains entirely unaddressed.

### 2.2 Skeleton-Based Motion Prediction

Predicting future poses from skeleton sequences is a core computer vision task with broad applications (Lyu et al., 2022). Early recurrent methods, including Residual-GRU (Martinez et al., 2017) and Structural-RNN (Jain et al., 2016), suffered from error accumulation and mean-pose collapse at longer horizons. Feed-forward approaches addressed this by predicting all frames simultaneously by LTD (Mao et al., 2019) and HRI (Mao et al., 2020) worked in the frequency domain with motion attention, while MST-GNN (Li et al., 2021) and MSR-GCN (Dang et al., 2021) improved spatial modeling via multi-scale graph convolutions. More recently, AuxFormer (Xu et al., 2023) used auxiliary tasks to strengthen Transformer representations, and siMLPe (Guo et al., 2023) showed that a simple MLP with DCT preprocessing can outperform complex architectures. Despite these, all existing methods operate on clean skeleton observations. To our knowledge, no prior work addresses motion prediction from WiFi CSI, a fundamentally harder setting that requires forecasting future poses from a noisy sensing modality. KOALA is the first to tackle this problem.

## 3 Methodology

### 3.1 Preliminaries

**Channel State Information.** Modern WiFi standards, such as IEEE 802.11, employ Orthogonal Frequency-Division Multiplexing (OFDM) to transmit data over multiple subcarriers simultaneously (792, 2017). The received signal on each subcarrier $s$ between transmit antenna $a_t$ and receive antenna $a_r$ is characterized by a complex-valued channel coefficient:

$$H(s, a_t, a_r) = |H(s, a_t, a_r)| e^{j\angle H(s, a_t, a_r)}, \tag{1}$$

where $|H|$ captures signal attenuation and $\angle H$ encodes the phase shift from multipath propagation. When a person moves, body segments reflect and scatter the signal along multiple paths, each contributing an

attenuated and phase-shifted copy:

$$H(s) \;=\; \sum_{l=1}^{L} \alpha_l \, e^{-j2\pi d_l/\lambda_s}, \tag{2}$$

where $\alpha_l$ and $d_l$ are the attenuation and propagation distance of path $l$, and $\lambda_s$ is the wavelength of sub-carrier $s$. As the body moves, path lengths $d_l$ change over time, inducing time-varying fluctuations in both $|H(s, a_t, a_r)|$ and $\angle H(s, a_t, a_r)$ across all subcarrier-antenna pairs that encode the body's spatial configuration. Stacking these coefficients over $T$ time steps, $S$ subcarriers, and $A$ antenna pairs yields the full CSI tensor $\mathbf{X} \in \mathbb{R}^{T \times S \times A}$, where each entry corresponds to $H(s, a_t, a_r)$ at a given time step. In KOALA, $\mathbf{X}$ serves both as direct contextual input for motion dynamics and as the sole sensory source for estimating observed poses, enabling fully end-to-end operation from raw WiFi signals.

**Koopman Operator Theory.** Human motion is inherently nonlinear, but Koopman theory (Koopman, 1931) provides a principled linearisation. Let $\mathbf{v}_t \in \mathcal{V}$ denote the body state at time $t$, representing the configuration of the human body in observation space. For a nonlinear body motion model $\mathbf{v}_{t+1} = F(\mathbf{v}_t)$ with transition $F$, the Koopman operator $\mathcal{K}$ acts on observables $g : \mathcal{V} \to \mathbb{R}^d$ via:

$$(\mathcal{K}g)(\mathbf{v}) \;=\; g\big(F(\mathbf{v})\big), \quad \forall \mathbf{v} \in \mathcal{V}. \tag{3}$$

Although $F$ is nonlinear, $\mathcal{K}$ is always linear (Koopman, 1931). For vector-valued observables $\boldsymbol{g} = [g_1, \ldots, g_N]^\top$, the lifted state evolves as $\boldsymbol{g}(\mathbf{v}_{t+1}) = \mathbf{K}\,\boldsymbol{g}(\mathbf{v}_t)$ with $\mathbf{K} \in \mathbb{R}^{N \times N}$, and multi-step prediction at horizon $h$ is $\boldsymbol{g}(\mathbf{v}_{t+h}) = \mathbf{K}^h \boldsymbol{g}(\mathbf{v}_t)$, requiring no autoregressive iteration. The main challenge is learning a lifting $\boldsymbol{g}$ where the finite-dimensional approximation remains accurate (Brunton & Kutz, 2019; Lusch et al., 2018). In KOALA, $N$ corresponds to the Koopman latent dimension $D_z$ and $\boldsymbol{g}$ is the composition of a skeleton-aware pose encoder, a dual-stream fusion module, and a lifting network, while $\mathbf{K}$ is CSI-conditioned in residual form, allowing the WiFi signal to modulate dynamics without the pose model encoding identity information.

### 3.2 Problem Formulation

We predict future human poses from WiFi CSI alone at inference. Let $\{\mathbf{x}_t\}_{t=1}^{T}$ with $\mathbf{x}_t \in \mathbb{R}^{D_c}$ denote $T$ CSI frames, where $D_c = S \times A$ is the product of the number of subcarriers $S$ and antenna pairs $A$. Here $J$ denotes the number of body joints and $\mathcal{H} = \{1, 3, 5, 10, 15, 20\}$ the set of prediction horizons in frames. Our goal is to predict poses $\hat{\mathbf{p}}_{T+h} \in \mathbb{R}^{J \times 3}$ at each horizon $h \in \mathcal{H}$:

$$\hat{\mathbf{p}}_{T+h} \;=\; f_\theta(\{\mathbf{x}_t\}_{t=1}^{T}), \quad h \in \mathcal{H}. \tag{4}$$

During training, ground-truth observed poses $\mathbf{P}_{1:T} = \{\mathbf{p}_t\}_{t=1}^{T}$ and future poses $\mathbf{P}_{T+1:T+\zeta}$ where $\zeta = \max(\mathcal{H})$ are available; at inference, only CSI is required. This task is challenging on three fronts: 1) CSI-to-pose mapping is noisy and environment-dependent due to multipath fading and subject-specific body geometry; 2) multi-horizon prediction requires temporal consistency, and autoregressive methods accumulate errors at long horizons; and 3) pose estimation errors propagate into the dynamics module, creating a cascaded-error problem absent from prior work that assumes clean pose input. We therefore seek a unified model that estimates observed poses from CSI, models their temporal dynamics via a Koopman operator with a residual parametrization that prevents identity collapse, and predicts across all horizons without error accumulation.

### 3.3 Approach Overview

As illustrated in Fig 2, KOALA operates in four sequential stages. *(i)* **CSI Encoding**: a Mamba-based encoder (Gu & Dao, 2024) extracts per-frame features $\{\mathbf{h}_t\}_{t=1}^{T}$ and a global context vector $\mathbf{c}$ via learned attention pooling. *(ii)* **Integrated Pose Estimation and Encoding**: a Mamba-based HPE module recovers estimated 3D poses from CSI features; a skeleton-aware encoder then converts those poses into compact structured features for the dynamics module. *(iii)* **Koopman Dynamics**: a dual-stream fusion combines CSI and pose features; a lifting network maps the result into a higher-dimensional Koopman latent space; a residual CSI-conditioned operator propagates the state across all prediction horizons in a single

rollout from the last observed latent state. *(iv)* **Anchor-Based Decoding**: the inverse lifting recovers per-horizon feature vectors from which an output head produces pose *deltas* relative to an anchor pose, and the final prediction is the sum of anchor and delta.

### 3.4 CSI Encoding

The CSI encoder processes $\mathbf{X} \in \mathbb{R}^{T \times D_c}$ into both per-frame features and a global context vector. Each frame is first projected by a two-layer MLP with GELU activation, then processed by $L_c$ stacked Mamba blocks with residual connections and layer normalization:

$$\mathbf{h}_t^{(0)} = \text{MLP}_{\text{proj}}(\mathbf{x}_t), \tag{5}$$

$$\mathbf{h}_t^{(\ell)} = \text{LN}\big(\text{Mamba}^{(\ell)}(\mathbf{h}_{1:T}^{(\ell-1)})_t + \mathbf{h}_t^{(\ell-1)}\big). \tag{6}$$

Mamba selective state space mechanism provides linear-time processing suited to the long-range correlations in CSI as the body moves through Fresnel zones. The per-frame representations $\mathbf{h}_{1:T}^{(L_c)}$ are shared downstream by both the HPE module and the dual-stream fusion. A learned attention pooling yields the global context vector:

$$\mathbf{c} = \sum_t \alpha_t \, \mathbf{h}_t^{(L_c)}, \qquad \alpha_t = \text{softmax}_t\big(\mathbf{w}^\top \mathbf{h}_t^{(L_c)}\big), \tag{7}$$

where $\mathbf{w} \in \mathbb{R}^d$ is learnable. The context $\mathbf{c}$ exclusively conditions the Koopman operator, implementing the inductive bias that CSI modulates *how* motion evolves rather than *what* the pose is, while avoiding quadratic cost over long sequences.

### 3.5 CSI Pose Estimation and Encoding

Raw CSI features are mapped to structured pose representations by two stages: a joint-conditioned Mamba HPE module that recovers 3D joint coordinates, followed by a skeleton-aware encoder that maps coordinates into kinematically structured features for the dynamics module. Both stages share the same learnable anatomical type embeddings $\{\mathbf{e}_j^{\text{type}}\}$ and skeleton adjacency bias $\mathbf{G}$, ensuring consistent joint identity and kinematic connectivity throughout. In both stages, the prediction loss is blocked from backpropagating into the HPE module via gradient detachment, isolating pose localization from the prediction objective.

**Stage 1: Joint-Conditioned HPE Module.** The HPE module converts per-frame CSI features $\{\mathbf{h}_t^{(L_c)}\}_{t=1}^T$ into estimated 3D poses $\{\tilde{\mathbf{p}}_t\}_{t=1}^T$, where $\tilde{\mathbf{p}}_t \in \mathbb{R}^{J \times 3}$. It is supervised during training by ground-truth observed poses $\{\mathbf{p}_t\}_{t=1}^T$ and used at inference as the sole source of pose information.

The module proceeds through four steps.

*Step 1: Temporal refinement.* A stack of $L_p$ Mamba blocks with residual connections and layer normalization refines the temporal context of the CSI features:

$$\mathbf{h}_t^{\text{hpe}} = \text{LN}\big(\text{Mamba}_{\text{hpe}}(\mathbf{h}_{1:T}^{(L_c)})_t + \mathbf{h}_t^{(L_c)}\big). \tag{8}$$

*Step 2: Joint token construction.* Each frame feature is projected and expanded into $J$ joint-specific tokens by adding a learnable anatomical embedding $\mathbf{e}_j^{\text{type}} \in \mathbb{R}^d$, which encodes the anatomical identity of joint $j$:

$$\mathbf{m}_{t,j} = \text{MLP}_{\text{expand}}\big(\mathbf{h}_t^{\text{hpe}}\big) + \mathbf{e}_j^{\text{type}}, \quad j = 1, \dots, J. \tag{9}$$

These embeddings $\{\mathbf{e}_j^{\text{type}}\}_{j=1}^J$ are shared with Stage 2, ensuring both stages reason about joints using the same anatomical vocabulary.

*Step 3: Per-joint temporal modeling.* Each joint's temporal sequence $\{\mathbf{m}_{t,j}\}_{t=1}^T$ is processed independently by a shared-weight Mamba block:

$$\mathbf{m}_{t,j}^{\text{out}} = \text{LN}\big(\text{Mamba}_{\text{joint}}(\mathbf{m}_{1:T,j})_t + \mathbf{m}_{t,j}\big). \tag{10}$$

Sharing weights across joints encourages learning of universal motion priors such as periodicity and velocity. The anatomical embeddings $\mathbf{e}_j^{\text{type}}$ retain joint-specific identity, so the model effectively uses the same temporal dynamics model for all joints but predicts their coordinates separately based on anatomical position.

*Step 4: Spatial self-attention and coordinate regression.* Within each frame, the $J$ joint tokens attend to one another via skeleton-biased multi-head self-attention. Let $\mathbf{M}_t = [\mathbf{m}_{t,1}^{\text{out}}, \ldots, \mathbf{m}_{t,J}^{\text{out}}]^{\top} \in \mathbb{R}^{J \times d}$ collect the $J$ joint tokens for frame $t$. The attention is modulated by a skeleton graph adjacency bias $\mathbf{G} \in \mathbb{R}^{J \times J}$ built from the human kinematic chain:

$$G_{ij} = \begin{cases} 0 & \text{if joints } i \text{ and } j \text{ are adjacent in the skeleton,} \\ -\beta & \text{otherwise,} \end{cases} \tag{11}$$

where $\beta > 0$ is a fixed suppression constant. The self-attention with multi-head projection matrices $\mathbf{W}_Q^{\text{hpe}}, \mathbf{W}_K^{\text{hpe}}, \mathbf{W}_V^{\text{hpe}} \in \mathbb{R}^{d \times d_h}$ where $d_h$ is the per-head dimension, is then:

$$\mathbf{M}_t^{\text{attn}} = \text{softmax}\left(\frac{(\mathbf{M}_t \mathbf{W}_Q^{\text{hpe}})(\mathbf{M}_t \mathbf{W}_K^{\text{hpe}})^{\top}}{\sqrt{d_h}} + \mathbf{G}\right) \mathbf{M}_t \mathbf{W}_V^{\text{hpe}}. \tag{12}$$

After a position-wise feed-forward sublayer and layer normalization, a per-joint two-layer MLP with GELU activation regresses the 3D coordinate of each joint:

$$\tilde{\mathbf{p}}_{t,j} = \text{MLP}_{\text{coord}}([\mathbf{M}_t^{\text{attn}}]_j) \in \mathbb{R}^3, \quad j = 1, \ldots, J, \tag{13}$$

where $[\cdot]_j$ denotes the $j$-th row. The full estimated pose is $\tilde{\mathbf{p}}_t = [\tilde{\mathbf{p}}_{t,1}, \ldots, \tilde{\mathbf{p}}_{t,J}] \in \mathbb{R}^{J \times 3}$.

**Stage 2: Skeleton-Aware Pose Encoding.** The skeleton encoder maps a pose sequence into a compact feature per frame that encodes kinematic relationships across joints, for consumption by the Koopman dynamics module. The estimated poses $\tilde{\mathbf{p}}_t$ are detached from the computation graph before entering this stage. This gradient barrier isolates the HPE module (trained solely on $\mathcal{L}_{\text{est}}$) from the prediction objective, preventing the prediction loss from adapting HPE outputs in ways that help downstream prediction at the expense of geometric pose accuracy. The skeleton encoder input follows a scheduled sampling protocol detailed in Appendix E.

*Spatial encoding.* Each joint coordinate $\tilde{\mathbf{p}}_{t,j} \in \mathbb{R}^3$ (or its ground-truth counterpart during teacher forcing) is embedded into $\mathbb{R}^d$ and combined with the shared anatomical type embedding:

$$\mathbf{e}_{t,j} = \text{MLP}_{\text{embed}}(\tilde{\mathbf{p}}_{t,j}) + \mathbf{e}_j^{\text{type}}, \quad j = 1, \ldots, J. \tag{14}$$

Let $\mathbf{E}_t = [\mathbf{e}_{t,1}, \ldots, \mathbf{e}_{t,J}]^{\top} \in \mathbb{R}^{J \times d}$, inter-joint dependencies are modelled via multi-head self-attention with the same adjacency bias $\mathbf{G}$ (equation 11), using the skeleton encoder's own projection matrices $\mathbf{W}_Q^{\text{skel}}, \mathbf{W}_K^{\text{skel}}, \mathbf{W}_V^{\text{skel}} \in \mathbb{R}^{d \times d_h}$, distinct from those of the HPE module:

$$\mathbf{E}_t^{\text{attn}} = \text{softmax}\left(\frac{(\mathbf{E}_t \mathbf{W}_Q^{\text{skel}})(\mathbf{E}_t \mathbf{W}_K^{\text{skel}})^{\top}}{\sqrt{d_h}} + \mathbf{G}\right) \mathbf{E}_t \mathbf{W}_V^{\text{skel}}. \tag{15}$$

After a position-wise feed-forward sublayer, the $J$ attended joint embeddings are concatenated and projected to a single per-frame representation:

$$\mathbf{f}_t^{\text{pose}} = \text{LN}\left(\mathbf{W}_{\text{pool}}[[\mathbf{E}_t^{\text{attn}}]_1; \ldots; [\mathbf{E}_t^{\text{attn}}]_J] + \mathbf{b}_{\text{pool}}\right) \in \mathbb{R}^d, \tag{16}$$

where $[;]$ denotes concatenation, $\mathbf{W}_{\text{pool}} \in \mathbb{R}^{d \times Jd}$, and $\mathbf{b}_{\text{pool}} \in \mathbb{R}^d$.

### 3.6 Dual-Stream Fusion

To ensure the Koopman operator receives informative input even when pose estimates are imperfect, we combine pose-derived features $\mathbf{f}_t^{\text{pose}}$ with CSI per-frame features $\mathbf{h}_t^{(L_c)}$ through additive residual fusion:

$$\mathbf{f}_t = \text{LN}\left(\mathbf{W}_c \mathbf{h}_t^{(L_c)} + \mathbf{W}_u \mathbf{f}_t^{\text{pose}} + \text{MLP}_{\text{fuse}}\left(\mathbf{W}_c \mathbf{h}_t^{(L_c)} + \mathbf{W}_u \mathbf{f}_t^{\text{pose}}\right)\right). \tag{17}$$

Additive fusion is intentional; it guarantees that the dynamics module receives CSI information directly, so that inaccurate pose estimates cannot completely dominate the representation. The contribution balance between the two streams is monitored during training via the ratio $\|\mathbf{W}_u \mathbf{f}_t^{\mathrm{pose}}\|_2 / \|\mathbf{W}_c \mathbf{h}_t^{(L_c)}\|_2$, providing a diagnostic of whether the skeleton stream contributes meaningfully throughout training. The fused sequence $\{\mathbf{f}_t\}_{t=1}^T$ is passed through $L_t$ Mamba blocks with residual connections, yielding temporally contextualised features $\{\tilde{\mathbf{f}}_t\}_{t=1}^T$ that capture motion dynamics for the subsequent Koopman lifting.

### 3.7 Koopman Lifting and Dynamics

**Lifting Network.** The lifting $\phi : \mathbb{R}^d \to \mathbb{R}^{D_z}$ maps temporally contextualized fused features into a higher-dimensional Koopman space where linear dynamics hold. Its inverse $\phi^{-1} : \mathbb{R}^{D_z} \to \mathbb{R}^d$ recovers feature-space representations. Both are three-layer MLPs with GELU, dropout, and layer normalization:

$$\mathbf{z}_t = \phi(\tilde{\mathbf{f}}_t) = \mathrm{LN}\big(\mathrm{MLP}_{\mathrm{enc}}(\tilde{\mathbf{f}}_t)\big) \in \mathbb{R}^{D_z}, \tag{18}$$

$$\hat{\tilde{\mathbf{f}}}_t = \phi^{-1}(\mathbf{z}_t) = \mathrm{LN}\big(\mathrm{MLP}_{\mathrm{dec}}(\mathbf{z}_t)\big) \in \mathbb{R}^d. \tag{19}$$

Setting $D_z > d$ is grounded in Koopman theory: the true operator is infinite-dimensional (Koopman, 1931), and any finite-dimensional approximation in $\mathbb{R}^{D_z}$ incurs a projection error that decreases as $D_z$ grows (Brunton & Kutz, 2019; Strässer et al., 2023). We set $D_z = 2d$, providing additional degrees of freedom for the learned eigenfunctions to disentangle nonlinear modes that are coupled in the original feature coordinates. FIX

*Remark* 1 (Lifting information preservation). If the reconstruction error $\|\phi^{-1}(\phi(\tilde{\mathbf{f}}_t)) - \tilde{\mathbf{f}}_t\|^2 \le \epsilon_r$ for all training features, then $\phi$ is approximately injective on the training manifold: for any two training features $\tilde{\mathbf{f}}_a, \tilde{\mathbf{f}}_b$,

$$\|\tilde{\mathbf{f}}_a - \tilde{\mathbf{f}}_b\| \le \|\phi^{-1}(\phi(\tilde{\mathbf{f}}_a)) - \phi^{-1}(\phi(\tilde{\mathbf{f}}_b))\| + 2\sqrt{\epsilon_r}. \tag{20}$$

In particular, if two feature vectors are separated by more than $2\sqrt{\epsilon_r}$, their lifted representations must be distinct (proof in Appendix A).

**Residual CSI-Conditioned Operator.** A fundamental difficulty in learning Koopman operators with gradient descent is the *identity attractor*. When a dense matrix $\mathbf{K}$ is initialized near identity, the prediction loss for slowly-varying motion is already low, so gradients provide little pressure to push $\mathbf{K}$ away from $\mathbf{I}$. Meanwhile, reconstruction and linearity losses actively penalize deviations from $\mathbf{K} \approx \mathbf{I}$, thereby keeping the operator uninformative throughout training.

KOALA resolves this by parametrizing the operator in **residual form**:

$$\mathbf{K}_{\mathrm{eff}} = \mathbf{I} + \mathbf{B} + \gamma\, \mathbf{U}(\mathbf{c})\, \mathbf{V}(\mathbf{c})^\top, \tag{21}$$

where $\mathbf{B} \in \mathbb{R}^{D_z \times D_z}$ is the learned residual dynamics matrix (we use $\mathbf{B}$ to distinguish it from the skeleton adjacency bias $\mathbf{G}$ of Section 3.5), $\mathbf{U}(\mathbf{c}), \mathbf{V}(\mathbf{c}) \in \mathbb{R}^{D_z \times r}$ are low-rank CSI adaptation factors produced by two-layer MLPs from the global context $\mathbf{c}$, and $\gamma > 0$ is a learnable scalar parametrised in log-space as $\gamma = \exp(\xi)$. The identity is baked into the architecture rather than enforced during initialization, so $\mathbf{B}$ faces no competing pressure to remain near zero. The log-space parametrization of $\gamma$ prevents the CSI adaptation from collapsing to zero as gradients weaken, keeping the WiFi conditioning active throughout training.

The rank constraint $r \ll D_z$ ensures CSI modulates a low-dimensional subspace of the dynamics, capturing environment-dependent multipath effects without dominating the base residual dynamics. The low-rank factors $\mathbf{U}(\mathbf{c})$ and $\mathbf{V}(\mathbf{c})$ are computed once per sample and reused across all rollout steps. The one-step update is:

$$\mathbf{z}_{t+1} = \mathbf{z}_t + \mathbf{B}\, \mathbf{z}_t + \gamma\, \mathbf{U}(\mathbf{c})\big(\mathbf{V}(\mathbf{c})^\top \mathbf{z}_t\big). \tag{22}$$

FIX

*Remark* 2 (Operator stability). Let $\mathbf{K}_{\mathrm{eff}}$ be defined as in equation 21 with $\|\mathbf{B}\|_F \le \eta_B$. Then $\|\mathbf{K}_{\mathrm{eff}}\|_2 \le 1 + \eta_B + \gamma \|\mathbf{U}\|_2 \|\mathbf{V}\|_2$, and for $h$-step rollouts $\|\mathbf{K}_{\mathrm{eff}}^h\|_2 \le (1 + \eta_B + \gamma \|\mathbf{U}\|_2 \|\mathbf{V}\|_2)^h$. With $\eta_B$ small at initialization and $\gamma \approx 0.1$, the amplification factor at $h = 20$ remains bounded during early training, preventing runaway gradient accumulation through long operator chains. The Frobenius norm $\|\mathbf{B}\|_F$ is monitored as a training diagnostic to track operator growth.

**Anchor-Based Multi-Horizon Prediction.** To predict at horizons $\mathcal{H}$, KOALA iterates the operator from the last lifted state $\mathbf{z}_T$:

$$\mathbf{z}_{T+k} = \mathbf{K}_{\text{eff}} \mathbf{z}_{T+k-1}, \quad k = 1, \ldots, \zeta. \tag{23}$$

Rather than decoding absolute poses, KOALA predicts pose *deltas* relative to an anchor pose $\bar{\mathbf{p}}$, the last observed pose during teacher forcing or the last estimated pose at inference. Denoting the delta head $\Delta_\theta = \text{MLP}_{\text{out}} \circ \phi^{-1}$:

$$\hat{\mathbf{p}}_{T+h} = \bar{\mathbf{p}} + \Delta_\theta(\mathbf{z}_{T+h}), \quad h \in \mathcal{H}. \tag{24}$$

This formulation eliminates a common degenerate solution: without the anchor, outputting the current pose at every horizon incurs nearly zero loss for slow motions. Under the anchor formulation, $\Delta_\theta(\mathbf{z}_{T+h}) \approx \mathbf{0}$ for all $h$ incurs proportional loss at long horizons where the true pose has drifted from the anchor, so the operator is forced to encode genuine temporal dynamics. The encoding pipeline executes once per sample; each additional prediction horizon costs only one matrix-vector product.

The following theorem formalizes why linear rollout in the Koopman space produces bounded multi-horizon predictions, while the same rollout in observation space does not.

**Theorem 1** (Bounded multi-horizon prediction). *Let $\mathbf{v}_{t+1} = F(\mathbf{v}_t)$ be the nonlinear dynamics governing human motion in observation space. Suppose the lifting $\phi$ satisfies the approximate Koopman invariance condition*

$$\|\phi(F(\mathbf{v})) - \mathbf{K}_{\text{eff}}\phi(\mathbf{v})\| \leq \varepsilon \quad \text{for all observed states } \mathbf{v},$$

*and that $\kappa := \|\mathbf{K}_{\text{eff}}\|_2 \leq 1 + \eta$ with $\eta \ll 1$ (an empirical condition monitored via $\|\mathbf{B}\|_F$ during training; see* Remark 2). *Let $L$ be the Lipschitz constant of the delta head $\Delta_\theta = \text{MLP}_{\text{out}} \circ \phi^{-1}$. Then the $h$-step prediction* FIX *error in observation space satisfies*

$$\|\hat{\mathbf{p}}_{T+h} - \mathbf{p}_{T+h}\| \leq L\frac{\kappa^h - 1}{\kappa - 1}\varepsilon + L\delta_h + e_{\text{dec},h} + \|\bar{\mathbf{p}} - \mathbf{p}_T\|, \tag{25}$$

*where $z^*T+h := \phi(F^h(v_T))$ is the ideal Koopman embedding of the true future state ($v_T \in \mathcal{V}$ as in Sec-* FIX *tion 3.1), and $\delta_h := |\phi(\tilde{f}T + h) - \phi(F^h(v_T))|$ measures how well the temporal-encoder feature at frame $T+h$ approximates this ideal embedding. The term $e_{\text{dec},h} := \|\mathbf{p}_T + \Delta_\theta(\mathbf{z}^*_{T+h}) - \mathbf{p}_{T+h}\|$ is the delta-head decoder approximation error: since $\Delta_\theta$ is a learned MLP rather than an exact inverse map, it does not perfectly reconstruct the true pose even when given the ideal future latent state $\mathbf{z}^*_{T+h}$ directly, and $e_{\text{dec},h}$ captures this residual, distinct from the invariance defects $\varepsilon$ and $\delta_h$ that arise upstream of the decoder. and $\|\bar{\mathbf{p}} - \mathbf{p}_T\|$ is the anchor alignment error (zero under teacher forcing). When $\kappa \leq 1 + \eta$ with small $\eta$, the geometric factor satisfies $(\kappa^h - 1)/(\kappa - 1) \leq h(1 + \eta h/2)$, yielding near-linear growth:*

$$\|\hat{\mathbf{p}}_{T+h} - \mathbf{p}_{T+h}\| \lesssim Lh\varepsilon(1 + \eta h/2) + L\delta_h + e_{\text{dec},h} + \|\bar{\mathbf{p}} - \mathbf{p}_T\|. \tag{26}$$

*In contrast, an autoregressive predictor iterating a decoder in observation space with per-step error $\varepsilon_{\text{ar}}$ incurs*

$$\|\hat{\mathbf{p}}^{\text{ar}}_{T+h} - \mathbf{p}_{T+h}\| \geq h\varepsilon_{\text{ar}} - O(h^2\varepsilon_{\text{ar}}^2). \tag{27}$$

*The key asymmetry is that $\varepsilon$ is measured in a latent space explicitly shaped to minimize Koopman invariance defect via the* KAL *objective (equation 31), whereas $\varepsilon_{\text{ar}}$ accumulates in the raw pose observation space where* FIX *no such regularisation applies. Consequently $\varepsilon \ll \varepsilon_{\text{ar}}$, and the bound tightens as $\mathcal{L}_{\text{KAL}}$ decreases during* FIX *training. (Proof and supporting remarks in Appendix B.)*

In practice, the lifting $\phi$ and operator $\mathbf{K}_{\text{eff}}$ are learned jointly from finite data, and the Koopman invariance condition $\phi(F(\mathbf{v})) = \mathbf{K}_{\text{eff}}\phi(\mathbf{v})$ holds only approximately, we denote this residual invariance defect by $\varepsilon$ and report it empirically for the trained model in Appendix I, rather than assuming it as an architectural guarantee.

### 3.8 Loss Design

Our training objective comprises three terms that jointly shape the Koopman latent space, supervise pose estimation, and enforce multi-horizon consistency.

**Prediction Loss.** The prediction loss penalizes pose-space error at each horizon with weights $w_h$ that emphasize long horizons:

$$\mathcal{L}_{\text{pred}} \;=\; \frac{1}{\sum_h w_h} \sum_{h \in \mathcal{H}} w_h \left\| \hat{\mathbf{p}}_{T+h} - \mathbf{p}_{T+h} \right\|_2^2. \tag{28}$$

Training simultaneously across all horizons enforces operator consistency: since multi-step predictions compose the same operator $\mathbf{K}_{\text{eff}}$, gradients from every horizon collectively constrain a single set of parameters.

**Koopman Anchored Latent Loss.** The KAL loss regularises the lifting and operator by requiring that operator rollouts match the temporal-encoder features the model would produce upon observing future frames. Operating in this feature space $\mathbb{R}^d$ rather than in raw observation or Koopman latent space decouples the supervision signal from CSI noise while maintaining gradient pressure through $\phi$ and $\mathbf{K}_{\text{eff}}$.    FIX

*Future feature targets.* Let $\mathbf{f}_{T+h}^* \in \mathbb{R}^d$ denote the temporal-encoder feature produced by processing the concatenated observation-and-future pose sequence through the fusion and temporal encoder. Under a no-gradient context, these targets are computed by a single extended forward pass:

$$\left[ \tilde{\mathbf{f}}_1, \ldots, \tilde{\mathbf{f}}_T, \mathbf{f}_{T+1}^*, \ldots, \mathbf{f}_{T+\zeta}^* \right] \;=\; \text{TemporalEnc}\Big( \text{Fusion}\big( \mathbf{h}_{1:T+\zeta}^{\text{pad}}, \, \mathbf{p}_{1:T+\zeta}^{\text{ext}} \big) \Big), \tag{29}$$

where $\mathbf{h}_{1:T+\zeta}^{\text{pad}}$ pads the last CSI memory frame for future positions (no future CSI is available), and $\mathbf{p}_{1:T+\zeta}^{\text{ext}} = [\mathbf{p}_{1:T}, \mathbf{p}_{T+1:T+\zeta}]$ uses ground-truth future poses. The extended forward pass is run under `torch.no_grad()`, so the targets carry no gradient. Also, ground-truth future poses $p_{T+1:T+\zeta}$ serve as supervision labels in    FIX
the same sense as $\mathcal{L}$pred, future CSI is never accessed, and the observation window $p1 : T$ follows the same scheduled sampling protocol as the main forward pass (Appendix E). We deliberately pad future CSI with the last observed frame $\mathbf{h}_T$ to match the information available at inference time, where no future CSI is accessible. The ground-truth future pose provides the dynamical evolution signal for supervision, while the static CSI maintains the environmental context. This design prevents the auxiliary loss from exploiting privileged future environmental information, ensuring consistency between training and inference conditions.

*Anchored KAL formulation.* Let $\mathbf{z}_T = \phi(\tilde{\mathbf{f}}_T)$ be the last lifted observation state, $\mathbf{r}_T = \phi^{-1}(\mathbf{z}_T)$ the no-rollout reconstruction, and $\tilde{\mathbf{f}}_{T+h}^{\text{pred}} = \phi^{-1}(\mathbf{K}_{\text{eff}}^h \mathbf{z}_T)$ the feature predicted by rolling out the operator $h$ steps. KAL measures how much the operator-predicted feature *changes* from the current reconstruction and compares this against the actual feature change:

$$\mathbf{f}_{T+h}^{\text{KAL}} \;=\; \tilde{\mathbf{f}}_T + \big( \tilde{\mathbf{f}}_{T+h}^{\text{pred}} - \mathbf{r}_T \big). \tag{30}$$

The KAL loss combines a $k=0$ reconstruction term with horizon-weighted $k>0$ prediction terms:

$$\mathcal{L}_{\text{KAL}} \;=\; \frac{1}{W} \left[ \| \mathbf{r}_T - \tilde{\mathbf{f}}_T \|_2^2 \;+\; \sum_{h \in \mathcal{H}} w_h^{\text{KAL}} \left\| \mathbf{f}_{T+h}^{\text{KAL}} - \mathbf{f}_{T+h}^* \right\|_2^2 \right], \tag{31}$$

where $w_h^{\text{KAL}} = 1/(1+h)$ and $W = 1 + \sum_h w_h^{\text{KAL}}$ is a normalising constant. The horizon weighting is intentionally *inverted* relative to $\mathcal{L}_{\text{pred}}$: short horizons dominate the KAL gradient signal. This is critical for stability, as the $h$-step operator chain $\mathbf{K}_{\text{eff}}^h$ amplifies gradient norms geometrically in $h$, so up-weighting long horizons in the KAL loss would produce unbounded gradients at initialization. Instead, $\mathcal{L}_{\text{pred}}$ handles long-horizon supervision while KAL provides stable gradient flow through the lifting and operator at all timescales.

The anchoring also eliminates the large-magnitude baseline at initialization. At initialisation $\mathbf{K}_{\text{eff}} \approx \mathbf{I}$, so $\tilde{\mathbf{f}}_{T+h}^{\text{pred}} \approx \mathbf{r}_T$ and $\mathbf{f}_{T+h}^{\text{KAL}} \approx \tilde{\mathbf{f}}_T$. The $k>0$ loss is then bounded by $\| \mathbf{f}_{T+h}^* - \tilde{\mathbf{f}}_T \|^2$, which is small because adjacent features differ little, preventing the gradient explosion through 20-step operator chains that occurs with an unanchored formulation.    NEW

The following proposition formalises this stability role of $\mathcal{L}_{\text{KAL}}$: removing it entirely admits a degenerate joint collapse of the lifting and reconstruction networks that leaves $\| \mathbf{B} \|_F$ unconstrained and causes the gradient of $\mathcal{L}_{\text{pred}}$ to diverge (proof in Appendix C).

Table 1: Performance comparison across prediction horizons on the WiPose dataset.

| Methods | Metrics | 100ms | 300ms | 500ms | 1000ms |
|---|---|---|---|---|---|
| MetaFi++ (Zhou et al., 2023) | MPJPE ↓ | 42.74 | 41.78 | 40.62 | 38.64 |
| | PA-MPJPE ↓ | 26.62 | 26.21 | 26.22 | 25.74 |
| HPE-Li (D. Gian et al., 2024) | MPJPE ↓ | 40.00 | 39.64 | 39.98 | 39.69 |
| | PA-MPJPE ↓ | 26.69 | 26.60 | 26.92 | 27.01 |
| ConvLSTM (Shi et al., 2015) | MPJPE ↓ | 66.44 | 65.91 | 66.02 | 66.57 |
| | PA-MPJPE ↓ | 37.00 | 36.96 | 36.87 | 37.15 |
| VMRNN (Tang et al., 2024) | MPJPE ↓ | 32.20 | 31.68 | 31.28 | 31.22 |
| | PA-MPJPE ↓ | 23.36 | 23.37 | 23.32 | 23.30 |
| LTD (Mao et al., 2019) | MPJPE ↓ | 42.07 | 41.00 | 40.38 | 39.35 |
| | PA-MPJPE ↓ | 28.57 | 28.41 | 28.44 | 28.38 |
| siMLPe (Guo et al., 2023) | MPJPE ↓ | 39.54 | 39.41 | 39.24 | 39.14 |
| | PA-MPJPE ↓ | 26.62 | 26.95 | 27.07 | 26.60 |
| **KOALA (Ours)** | MPJPE ↓ | 26.14 | 26.03 | 26.49 | 27.28 |
| | PA-MPJPE ↓ | 20.81 | 20.77 | 20.98 | 21.28 |

**Proposition 1** (Necessity of $\mathcal{L}_{\mathrm{KAL}}$). *Suppose $\mathcal{L}_{\mathrm{KAL}}$ is removed entirely. Then $\mathcal{L}_{\mathrm{pred}} + \lambda_e \mathcal{L}_{\mathrm{est}}$ admits a degenerate joint minimiser in which $\phi$ and $\phi^{-1}$ co-adapt so that $\phi$ maps all features to an $\varepsilon$-neighbourhood of the origin while $\phi^{-1}$ compensates by scaling up, leaving $\hat{\mathbf{p}}_{T+h}$ unchanged for any $\varepsilon > 0$. Under this collapse $\|\mathbf{B}\|_F$ is unconstrained, and the gradient of $\mathcal{L}_{\mathrm{pred}}$ with respect to $\mathbf{B}$ satisfies*

$$\left\| \nabla_{\mathbf{B}} \, \mathcal{L}_{\mathrm{pred}}^{(h)} \right\|_F \; \leq \; 2 \, L_\Delta \cdot h \cdot \|\mathbf{K}_{\mathrm{eff}}\|_2^{h-1} \cdot \|\mathbf{z}_T\| \cdot \|\hat{\mathbf{p}}_{T+h} - \mathbf{p}_{T+h}\|, \tag{32}$$

*where $L_\Delta$ is the Lipschitz constant of the delta head $\Delta_\theta = \mathrm{MLP}_{\mathrm{out}} \circ \phi^{-1}$. Since $\|\mathbf{K}_{\mathrm{eff}}\|_2 \to \infty$ as $\|\mathbf{B}\|_F \to \infty$, the bound grows as $\mathcal{O}(\|\mathbf{K}_{\mathrm{eff}}\|_2^{h-1})$, diverging at $h = 20$ as $\|\mathbf{B}\|_F^{19}$. $\mathcal{L}_{\mathrm{KAL}}$ prevents this collapse: if $\mathbf{z}_T \approx \mathbf{0}$ then $\phi^{-1}(\mathbf{K}_{\mathrm{eff}}^h \mathbf{z}_T) \approx \phi^{-1}(\mathbf{0})$, a constant independent of $h$, which cannot match $f_{T+h}^*$ for all $h \in \mathcal{H}$ simultaneously since target features vary with the subject motion.*

**CSI Pose Estimation Loss.** The HPE module is supervised with ground-truth observed poses:

$$\mathcal{L}_{\mathrm{est}} \; = \; \frac{1}{T} \sum_{t=1}^{T} \|\tilde{\mathbf{p}}_t - \mathbf{p}_t\|_2^2. \tag{33}$$

This is the sole training signal for the HPE module; because estimated poses are detached before entering the skeleton encoder, the prediction loss cannot bias the HPE module towards features that help prediction at the cost of localization accuracy.

**Total Loss.** The total loss function can be expressed by the following:

$$\mathcal{L} \; = \; \lambda_p \, \mathcal{L}_{\mathrm{pred}} \; + \; \lambda_k \, \mathcal{L}_{\mathrm{KAL}} \; + \; \lambda_e \, \mathcal{L}_{\mathrm{est}}, \tag{34}$$

with $\lambda_p = 1.5$, $\lambda_k = 0.5$, $\lambda_e = 1.0$. This three-term objective is more compact than the seven-term objectives used in prior Koopman-based motion prediction methods, which required separate linearity, contrastive, reconstruction, auxiliary, and spectral-stability losses. The KAL loss subsumes reconstruction (the $k=0$ term), Koopman linearity (via feature-space prediction), and the stability function previously served by InfoNCE, without requiring batch-negative mining or spectral norm estimation at each training step.

Table 2: Evaluation performance across protocols and settings at different horizons (ms) on the MM-Fi.

| Setting | Metric | Protocol 1 | | | | | | Protocol 2 | | | | | | Protocol 3 | | | | | |
|---|---|---|---|---|---|---|---|---|---|---|---|---|---|---|---|---|---|---|---|
| | | 100 | 300 | 500 | 1000 | 1500 | 2000 | 100 | 300 | 500 | 1000 | 1500 | 2000 | 100 | 300 | 500 | 1000 | 1500 | 2000 |
| S1 | MPJPE ↓ | 54.5 | 58.2 | 62.5 | 66.7 | 65.0 | 63.6 | 53.2 | 56.9 | 60.8 | 64.0 | 62.1 | 61.0 | 55.2 | 62.3 | 69.2 | 76.2 | 73.3 | 69.8 |
| | PA-MPJPE ↓ | 45.5 | 48.0 | 51.2 | 54.1 | 53.4 | 52.6 | 44.3 | 46.6 | 49.1 | 51.0 | 50.2 | 49.8 | 46.5 | 52.6 | 58.3 | 61.3 | 63.2 | 60.1 |
| | PCK@20 ↑ | 91.0 | 89.5 | 87.6 | 85.6 | 86.5 | 87.3 | 91.3 | 89.9 | 88.2 | 86.6 | 87.5 | 88.3 | 90.6 | 87.8 | 85.0 | 82.3 | 83.7 | 85.3 |
| | PCK@10 ↑ | 72.5 | 69.9 | 67.3 | 64.7 | 65.9 | 67.1 | 73.1 | 70.6 | 68.1 | 66.0 | 67.2 | 68.3 | 72.0 | 68.3 | 65.1 | 62.3 | 64.1 | 65.8 |
| S2 | MPJPE ↓ | 55.7 | 62.0 | 68.4 | 74.7 | 71.9 | 68.7 | 52.8 | 58.2 | 63.5 | 68.0 | 65.4 | 63.5 | 53.1 | 59.0 | 64.5 | 68.9 | 66.4 | 64.4 |
| | PA-MPJPE ↓ | 47.3 | 52.5 | 57.7 | 62.8 | 61.8 | 59.2 | 44.0 | 48.0 | 51.8 | 54.7 | 53.5 | 52.6 | 44.7 | 49.0 | 52.9 | 55.5 | 54.4 | 53.5 |
| | PCK@20 ↑ | 90.3 | 87.8 | 85.2 | 82.6 | 84.1 | 85.6 | 91.5 | 89.4 | 87.2 | 85.2 | 86.3 | 87.4 | 91.3 | 89.1 | 86.7 | 84.6 | 85.8 | 86.9 |
| | PCK@10 ↑ | 71.7 | 68.5 | 65.3 | 62.6 | 61.4 | 66.2 | 73.5 | 70.1 | 67.1 | 64.2 | 65.9 | 67.7 | 73.4 | 69.8 | 66.7 | 63.8 | 65.5 | 67.2 |
| S3 | MPJPE ↓ | 52.1 | 57.0 | 61.9 | 65.7 | 63.4 | 61.9 | 51.5 | 54.6 | 58.5 | 62.2 | 60.6 | 59.4 | 52.6 | 58.3 | 63.2 | 67.2 | 64.5 | 62.5 |
| | PA-MPJPE ↓ | 43.5 | 47.0 | 50.5 | 52.8 | 51.8 | 51.1 | 42.8 | 44.9 | 47.6 | 50.0 | 49.2 | 48.9 | 41.2 | 48.5 | 52.0 | 54.3 | 52.9 | 51.7 |
| | PCK@20 ↑ | 91.8 | 89.9 | 87.8 | 85.9 | 86.9 | 87.9 | 92.0 | 90.9 | 89.3 | 87.6 | 88.3 | 88.9 | 91.6 | 89.3 | 87.2 | 85.4 | 86.5 | 87.7 |
| | PCK@10 ↑ | 73.9 | 70.7 | 67.8 | 65.1 | 66.5 | 67.7 | 74.3 | 72.1 | 69.6 | 67.0 | 68.1 | 69.5 | 73.7 | 70.2 | 67.3 | 64.6 | 66.3 | 67.8 |

## 4 Experiments

### 4.1 Datasets

We evaluate on two WiFi-based datasets: MM-Fi (Yang et al., 2023b) and WiPose (Zhou et al., 2022). MM-Fi provides large-scale multimodal data with synchronized CSI streams across 40 subjects, 27 actions, and 4 environments under three protocols (P1: daily activities; P2: rehabilitation exercises; P3: all 27 actions) and three settings (S1: random split; S2: cross-subject; S3: cross-environment). WiPose provides fine-grained CSI data with skeletal annotations for 2D pose understanding. Following the standard evaluation protocols, we evaluate on MPJPE and PA-MPJPE in millimeters, alongside PCKs of 20% and 10%.

### 4.2 2D Motion Prediction

Table 1 reports results on the WiPose dataset. KOALA achieves the lowest MPJPE and PA-MPJPE at every horizon, reaching 26.14 mm at 100 ms and 27.28 mm at 1000 ms, an improvement of approximately 19% over the strongest baseline VMRNN. Notably, KOALA degradation across horizons is only 1.14 mm in MPJPE, showing that the Koopman operator propagates latent states without meaningful error accumulation. Among baselines, ConvLSTM performs considerably worse, confirming that naive spatiotemporal convolution struggles in the lower-dimensional 2D pose space, while trajectory-based methods (LTD, siMLPe) and CSI-specialized architectures (MetaFi++, HPE-Li) cluster in the 39-42 mm range, unable to close the gap with state-space approaches.

### 4.3 3D Motion Prediction

Table 2 reports KOALA on MM-Fi across three protocols, three settings, and six horizons. MPJPE grows moderately with horizon, with PA-MPJPE below 65.3 mm and PCK@20 above 82.3% across all configurations. S3 achieves the lowest short-horizon errors (52.1 mm under P1 at 100 ms), while S2 is the most challenging (74.7 mm at 1000 ms under P1) due to unseen body geometries. Protocol 2 yields the lowest errors under S2 and S3 owing to its constrained action set, whereas Protocol 3 maintains PCK@20 above 85.4% at 1000 ms despite covering all 27 categories. Figure 3 confirms coherent rollouts with anatomically plausible skeletons at all horizons (qualitative results in Appendix F; ablations in Appendix J).

Table 3 compares KOALA against nine baselines on MM-Fi, spanning WiFi pose estimation methods (HPE-Li, MetaFi++), spatiotemporal prediction models (ConvLSTM, MIM, SwinLSTM, VMRNN), and skeleton-based methods (LTD, siMLPe, AuxFormer). KOALA outperforms all baselines by a substantial margin across every metric and horizon: at 100 ms it achieves 52.1 mm MPJPE, roughly 6.4× lower than the best competing method SwinLSTM (332.6 mm) and 73.9% PCK@10 while no baseline exceeds 18.1%. Among WiFi baselines, MetaFi++ consistently outperforms HPE-Li, confirming that stronger pose backbones yield more temporally stable estimates. Skeleton-based methods perform poorly under CSI-derived pose noise, whereas spatiotemporal models exhibit near-flat error profiles across horizons, suggesting mean regression

Table 3: Performance comparison across prediction horizons on the MM-Fi dataset.

| Methods | Metrics | 100ms | 300ms | 500ms | 1000ms | 1500ms | 2000ms |
|---|---|---|---|---|---|---|---|
| HPE-Li (D. Gian et al., 2024) | MPJPE ↓ | 375.2 | 383.4 | 351.6 | 351.2 | 339.8 | 368.5 |
| | PA-MPJPE ↓ | 120.7 | 122.3 | 119.4 | 118.4 | 119.5 | 122.8 |
| | PCK@20 ↑ | 3.8 | 4.3 | 7.0 | 8.6 | 10.0 | 6.3 |
| | PCK@10 ↑ | 0.3 | 0.2 | 0.8 | 1.0 | 1.1 | 0.5 |
| MetaFi++ (Zhou et al., 2023) | MPJPE ↓ | 311.7 | 308.4 | 311.2 | 312.8 | 316.9 | 312.0 |
| | PA-MPJPE ↓ | 105.5 | 105.4 | 104.6 | 104.7 | 105.1 | 105.1 |
| | PCK@20 ↑ | 78.8 | 80.7 | 78.4 | 78.4 | 75.9 | 80.1 |
| | PCK@10 ↑ | 18.1 | 17.7 | 18.4 | 17.2 | 18.2 | 16.8 |
| ConvLSTM (Shi et al., 2015) | MPJPE ↓ | 345.0 | 340.9 | 338.7 | 334.9 | 334.3 | 342.1 |
| | PA-MPJPE ↓ | 102.9 | 102.8 | 102.7 | 102.6 | 102.8 | 102.5 |
| | PCK@20 ↑ | 44.78 | 47.04 | 49.17 | 51.74 | 51.71 | 46.89 |
| | PCK@10 ↑ | 3.7 | 4.2 | 4.7 | 5.5 | 5.7 | 4.14 |
| MIM (Wang et al., 2019) | MPJPE ↓ | 335.6 | 337.5 | 340.1 | 339.8 | 338.5 | 337.3 |
| | PA-MPJPE ↓ | 102.9 | 102.8 | 102.6 | 102.7 | 102.5 | 102.1 |
| | PCK@20 ↑ | 53.85 | 52.75 | 49.87 | 49.44 | 51.64 | 52.90 |
| | PCK@10 ↑ | 6.2 | 5.7 | 5.4 | 5.20 | 5.9 | 5.8 |
| SwinLSTM (Tang et al., 2023) | MPJPE ↓ | 332.6 | 330.8 | 330.9 | 335.5 | 334.70 | 340.8 |
| | PA-MPJPE ↓ | 101.7 | 101.8 | 101.9 | 101.8 | 101.8 | 102.3 |
| | PCK@20 ↑ | 53.4 | 53.3 | 53.8 | 51.1 | 51.8 | 49.4 |
| | PCK@10 ↑ | 5.7 | 5.8 | 5.72 | 4.9 | 5.4 | 4.7 |
| VMRNN (Tang et al., 2024) | MPJPE ↓ | 338.2 | 340.1 | 343.1 | 349.9 | 343.8 | 333.7 |
| | PA-MPJPE ↓ | 102.6 | 101.9 | 101.9 | 102.0 | 101.8 | 102.1 |
| | PCK@20 ↑ | 48.1 | 48.7 | 47.5 | 43.9 | 48.5 | 52.4 |
| | PCK@10 ↑ | 7.2 | 7.8 | 7.6 | 7.4 | 7.9 | 8.3 |
| LTD (Mao et al., 2019) | MPJPE ↓ | 365.4 | 362.7 | 359.6 | 364.7 | 368.5 | 368.9 |
| | PA-MPJPE ↓ | 107.99 | 108.19 | 108.41 | 108.13 | 107.94 | 107.88 |
| | PCK@20 ↑ | 34.3 | 35.3 | 36.5 | 32.92 | 32.1 | 33.6 |
| | PCK@10 ↑ | 2.41 | 2.72 | 3.06 | 2.40 | 2.23 | 2.47 |
| siMLPe (Guo et al., 2023) | MPJPE ↓ | 348.7 | 357.1 | 365.7 | 376.9 | 383.6 | 380.4 |
| | PA-MPJPE ↓ | 106.3 | 105.8 | 104.2 | 107.4 | 107.4 | 108.2 |
| | PCK@20 ↑ | 28.34 | 28.25 | 25.07 | 20.75 | 18.74 | 20.14 |
| | PCK@10 ↑ | 2.82 | 3.46 | 3.35 | 2.98 | 2.61 | 2.10 |
| AuxFormer (Xu et al., 2023) | MPJPE ↓ | 373.2 | 385.9 | 382.5 | 357.2 | 394.5 | 367.6 |
| | PA-MPJPE ↓ | 103.6 | 101.8 | 102.2 | 102.8 | 101.4 | 101.49 |
| | PCK@20 ↑ | 18.6 | 15.5 | 18.5 | 24.75 | 11.74 | 21.75 |
| | PCK@10 ↑ | 1.53 | 1.89 | 1.75 | 2.90 | 1.06 | 3.07 |
| **KOALA (Ours)** | MPJPE ↓ | 52.1 | 57.0 | 61.9 | 65.7 | 63.4 | 61.9 |
| | PA-MPJPE ↓ | 43.5 | 47.0 | 50.5 | 52.8 | 51.8 | 51.1 |
| | PCK@20 ↑ | 91.8 | 89.9 | 87.8 | 85.9 | 86.9 | 87.9 |
| | PCK@10 ↑ | 73.9 | 70.7 | 67.8 | 65.1 | 66.5 | 67.7 |

rather than genuine temporal modeling. KOALA shows a generally increasing error trend with horizon, FIX reflecting genuine temporal modelling rather than static averaging.

## 5 Conclusion

We presented KOALA, a novel framework for anticipatory human motion prediction directly from WiFi CSI. By lifting noisy CSI-derived pose sequences into a learned Koopman latent space, KOALA achieves multi-

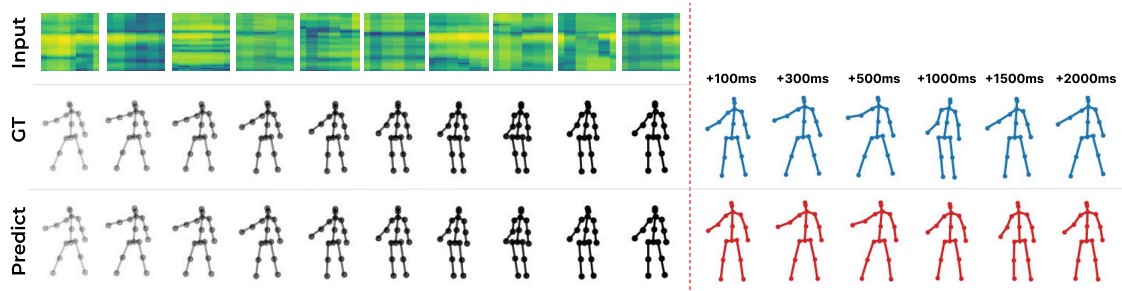

Figure 3: Qualitative results between predicted poses and Ground Truth (GT) for different time steps.

horizon prediction through simple matrix-vector products without autoregressive iteration. The residual parametrization resolves the identity attractor, the anchor-delta head eliminates mean-pose collapse, and the KAL loss enforces dynamical consistency without contrastive, spectral, or auxiliary losses. Experiments on MM-Fi and WiPose demonstrate substantial improvements over all baselines, with MPJPE growing moderately across horizons rather than accumulating exponentially as in autoregressive approaches. The current single-person formulation does not address signal superposition in multi-occupant settings. The CSI-conditioned operator partially accounts for environment variation but has not been validated under zero-shot cross-site transfer. We acknowledge that KOALA produces a single deterministic prediction per horizon and cannot capture the full distribution of possible future motions. Future work could extend KOALA to probabilistic prediction by modeling the distribution in the Koopman latent space. Finally, the Koopman assumption of approximately time-invariant dynamics within the observation window may break down at abrupt action transitions, a direction that is addressable through mixture-of-operators or mode-switching extensions.

**Broader Impact Statement**

This work contributes to the development of machine learning techniques for modeling temporal dynamics in wireless sensing data. The proposed approach has potential applications in areas such as healthcare monitoring and intelligent environments, while providing a more privacy-conscious alternative to vision-based sensing systems.

Similar to other sensing technologies, misuse or inappropriate deployment may raise concerns regarding privacy, user consent, and surveillance. However, these considerations are not unique to our approach and are broadly applicable across wireless and machine-learning-based sensing systems.

Overall, we do not anticipate any significant societal risks beyond those commonly associated with existing sensing and machine learning technologies.

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

## APPENDIX

## A    Proof of Remark 1

*Proof.* Fix any two training features $\tilde{\mathbf{f}}_a, \tilde{\mathbf{f}}_b \in \mathbb{R}^d$. By the triangle inequality:

$$\|\tilde{\mathbf{f}}_a - \tilde{\mathbf{f}}_b\| \leq \|\tilde{\mathbf{f}}_a - \phi^{-1}(\phi(\tilde{\mathbf{f}}_a))\| + \|\phi^{-1}(\phi(\tilde{\mathbf{f}}_a)) - \phi^{-1}(\phi(\tilde{\mathbf{f}}_b))\| + \|\phi^{-1}(\phi(\tilde{\mathbf{f}}_b)) - \tilde{\mathbf{f}}_b\|.$$

The first and third terms are individual round-trip reconstruction errors. Since $\mathcal{L}_{\text{rec}} \leq \epsilon_r$ implies each round-trip error is at most $\sqrt{\epsilon_r}$ on the training manifold, both terms are bounded by $\sqrt{\epsilon_r}$, giving:

$$\|\tilde{\mathbf{f}}_a - \tilde{\mathbf{f}}_b\| \leq \|\phi^{-1}(\phi(\tilde{\mathbf{f}}_a)) - \phi^{-1}(\phi(\tilde{\mathbf{f}}_b))\| + 2\sqrt{\epsilon_r}. \tag{35}$$

Therefore, whenever $\|\tilde{\mathbf{f}}_a - \tilde{\mathbf{f}}_b\| > 2\sqrt{\epsilon_r}$, the middle term must be strictly positive, which implies $\phi(\tilde{\mathbf{f}}_a) \neq \phi(\tilde{\mathbf{f}}_b)$. Hence $\phi$ is approximately injective on the training manifold, with injectivity guaranteed for any pair separated by more than $2\sqrt{\epsilon_r}$. □

## B    Proof of Theorem 1

FIX

*Proof.* Let the latent prediction error after $h$ Koopman transitions be decomposed as

$$\mathbf{K}_{\text{eff}}^h \mathbf{z}_T - \mathbf{z}_{T+h}^* = \left( \mathbf{K}_{\text{eff}}^h \mathbf{z}_T - \mathbf{z}_{T+h} \right) + \left( \mathbf{z}_{T+h} - \mathbf{z}_{T+h}^* \right), \tag{36}$$

where $\mathbf{z}_{T+h}^* = \phi(F^h(v_T))$ denotes the ideal Koopman embedding of the true future state.

The first term corresponds to the accumulated Koopman invariance defect. By applying the standard telescoping expansion,

$$\mathbf{K}_{\text{eff}}^h \mathbf{z}_T - \mathbf{z}_{T+h} = \sum_{k=0}^{h-1} \mathbf{K}_{\text{eff}}^{h-1-k} \left( \mathbf{K}_{\text{eff}} \mathbf{z}_{T+k} - \mathbf{z}_{T+k+1} \right). \tag{37}$$

Taking the norm and using the triangle inequality gives

$$\begin{aligned} \left\| \mathbf{K}_{\text{eff}}^h \mathbf{z}_T - \mathbf{z}_{T+h} \right\| &\leq \sum_{k=0}^{h-1} \|\mathbf{K}_{\text{eff}}\|_2^{h-1-k} \|\mathbf{K}_{\text{eff}} \mathbf{z}_{T+k} - \mathbf{z}_{T+k+1}\| \\ &\leq \sum_{k=0}^{h-1} \kappa^{h-1-k} \epsilon \\ &= \frac{\kappa^h - 1}{\kappa - 1} \epsilon, \end{aligned} \tag{38}$$

where $\kappa = \|\mathbf{K}_{\text{eff}}\|_2$ and $\epsilon$ denotes the one-step Koopman invariance defect.

We next map the latent error into the pose space. The predicted pose is generated by the anchor-based delta head:

$$\hat{\mathbf{p}}_{T+h} = \bar{\mathbf{p}} + \Delta_\theta(\mathbf{z}_{T+h}), \tag{39}$$

while the ideal prediction, obtained by applying the same decoder to the true future latent state $\mathbf{z}_{T+h}^*$, is

$$\mathbf{p}_{T+h} = \bar{\mathbf{p}} + \Delta_\theta(\mathbf{z}_{T+h}^*) + e_{\text{dec},h}, \tag{40}$$

where $e_{\mathrm{dec},h} := \mathbf{p}_{T+h} - \bar{\mathbf{p}} - \Delta_\theta(\mathbf{z}^*_{T+h})$ is the decoder approximation error, since $\Delta_\theta$ is a learned MLP and not an exact inverse map, it need not perfectly reconstruct $\mathbf{p}_{T+h}$ even when supplied with the ideal latent state $\mathbf{z}^*_{T+h}$ directly. Unlike $\varepsilon$ and $\delta_h$, which arise upstream of the decoder (in the invariance of the lifting and operator), $e_{\mathrm{dec},h}$ is a property of $\Delta_\theta$ itself and is not controlled by the Koopman machinery, we therefore retain it as an explicit, separately-reported term rather than bounding it by the anchor alignment error.

Since the anchor term is shared by both expressions, it cancels out. Using the triangle inequality,

$$\|\hat{\mathbf{p}}_{T+h} - \mathbf{p}_{T+h}\| \le \left\|\Delta_\theta(\mathbf{z}_{T+h}) - \Delta_\theta(\mathbf{z}^*_{T+h})\right\| + \|e_{\mathrm{dec},h}\|. \tag{41}$$

Assuming that $\Delta_\theta$ is Lipschitz continuous with constant $L$,

$$\left\|\Delta_\theta(\mathbf{z}_{T+h}) - \Delta_\theta(\mathbf{z}^*_{T+h})\right\| \le L\left\|\mathbf{z}_{T+h} - \mathbf{z}^*_{T+h}\right\|, \tag{42}$$

which gives the latent-to-pose error propagation term $L\delta_h$. Combining the above results yields

$$\|\hat{\mathbf{p}}_{T+h} - \mathbf{p}_{T+h}\| \le L\frac{\kappa^h - 1}{\kappa - 1}\epsilon + L\delta_h + e_{\mathrm{dec},h}. \tag{43}$$

Including the anchor alignment error $\|\bar{\mathbf{p}} - \mathbf{p}_T\|$ (zero under teacher forcing) as an additional, independent additive term gives the stated bound. We emphasize that $e_{\mathrm{dec},h}$ and $\|\bar{\mathbf{p}} - \mathbf{p}_T\|$ are logically distinct quantities: the former reflects the decoder's approximation quality given a perfect latent input, the latter reflects how far the anchor pose has drifted from the true current pose; conflating the two would understate the decoder contribution to the overall bound. When $\kappa = 1 + \eta$ and $\eta h$ is sufficiently small, the geometric accumulation term satisfies $(\kappa^h - 1)/(\kappa - 1) \lesssim h$. For the autoregressive lower bound in equation 27, each step feeds its own noisy prediction back as input, so errors accumulate additively at leading order with no latent-space regularisation to suppress $\varepsilon_{\mathrm{ar}}$. $\qquad\square$

*Remark* 3. The condition $\kappa \le 1 + \eta$ is an empirical property of the trained operator rather than an architectural guarantee. Unlike formulations that enforce a hard spectral constraint via projected gradient or power-iteration penalties, KOALA monitors $\|\mathbf{B}\|_F$ as a diagnostic and relies on the KAL loss and residual initialization to keep $\kappa$ near 1. This is sufficient in practice, as training curves show $\|\mathbf{B}\|_F$ remaining moderate throughout, but the bound in equation 25 should be understood as conditioned on this empirical regularity rather than as a hard guarantee.

## C   Proof of Proposition 1

NEW

*Proof.* **Step 1: Degenerate joint minimiser.** Fix $\varepsilon > 0$. Define $\phi_\varepsilon(\tilde{f}) = \varepsilon\,\bar{\phi}(\tilde{f})$ where $\bar{\phi}$ is a fixed injective encoder, and let $\phi_\varepsilon^{-1}$ be the corresponding left inverse scaled by $\varepsilon^{-1}$: $\phi_\varepsilon^{-1}(\mathbf{z}) = \bar{\phi}^{-1}(\mathbf{z}/\varepsilon)$. Both are realisable by the MLP lifting network. Since $\mathcal{L}_{\mathrm{KAL}}$ contains the reconstruction term $\|r_T - \tilde{f}_T\|^2 = \|\phi^{-1}(\phi(\tilde{f}_T)) - \tilde{f}_T\|^2$, once $\mathcal{L}_{\mathrm{KAL}}$ is removed the only remaining constraint on $\phi$ and $\phi^{-1}$ is that their composition approximates the identity, which holds for any $\varepsilon$:

$$\phi_\varepsilon^{-1}(\phi_\varepsilon(\tilde{f})) = \bar{\phi}^{-1}(\bar{\phi}(\tilde{f})) \approx \tilde{f}.$$

Hence $\phi$ and $\phi^{-1}$ co-adapt during training to jointly satisfy $\phi^{-1} \circ \phi \approx \mathrm{Id}$ for any encoder scale $\varepsilon$, because $\mathcal{L}_{\mathrm{est}}$ does not involve $\phi$ or $\phi^{-1}$ directly.

**Step 2: Prediction invariance to $\varepsilon$.** Under $\phi_\varepsilon$, $\mathbf{z}_T = \varepsilon\,\bar{\phi}(\tilde{f}_T)$ and $\mathbf{K}_{\mathrm{eff}}^h \mathbf{z}_T = \varepsilon\,\mathbf{K}_{\mathrm{eff}}^h \bar{\phi}(\tilde{f}_T)$. Since $\phi_\varepsilon^{-1}(\varepsilon\,\mathbf{u}) = \bar{\phi}^{-1}(\mathbf{u})$ for any $\mathbf{u}$, the prediction

$$\hat{\mathbf{p}}_{T+h} = \bar{\mathbf{p}} + \Delta_\theta\big(\bar{\phi}^{-1}(\mathbf{K}_{\mathrm{eff}}^h \bar{\phi}(\tilde{f}_T))\big)$$

is independent of $\varepsilon$, so $\mathcal{L}_{\mathrm{pred}}$ provides no gradient opposing $\varepsilon \to 0$ and $\|\mathbf{B}\|_F$ is unconstrained.

**Step 3: Gradient explosion.** Since $\mathbf{K}_{\mathrm{eff}} = \mathbf{I} + \mathbf{B} + \gamma \mathbf{U}\mathbf{V}^\top$, $\partial \mathbf{K}_{\mathrm{eff}}/\partial \mathbf{B}$ is the identity super-operator. By the product rule for matrix powers:

$$\frac{\partial}{\partial \mathbf{B}}\big(\mathbf{K}_{\mathrm{eff}}^h \mathbf{z}_T\big) = \sum_{k=0}^{h-1} \mathbf{K}_{\mathrm{eff}}^{h-1-k}\big(\mathbf{z}_T\,\mathbf{e}^\top\big)\mathbf{K}_{\mathrm{eff}}^k,$$

for unit perturbation direction $\mathbf{e}$. Sub-multiplicativity gives

$$\left\|\frac{\partial}{\partial \mathbf{B}}\mathbf{K}_{\mathrm{eff}}^h \mathbf{z}_T\right\|_F \leq h\,\|\mathbf{K}_{\mathrm{eff}}\|_2^{h-1}\,\|\mathbf{z}_T\|.$$

Composing with $\Delta_\theta$ ($L_\Delta$-Lipschitz, as it is a composition of Lipschitz MLPs) and applying the chain rule through $\mathcal{L}_{\mathrm{pred}}^{(h)} = \|\hat{\mathbf{p}}_{T+h} - \mathbf{p}_{T+h}\|^2$ yields equation equation 32. As $\|\mathbf{B}\|_F \to \infty$, sub-additivity gives $\|\mathbf{K}_{\mathrm{eff}}\|_2 \geq \|\mathbf{B}\|_2 - 1 - \gamma\|\mathbf{U}\|_2\|\mathbf{V}\|_2$, and since $\|\mathbf{B}\|_2 \geq \|\mathbf{B}\|_F/\sqrt{D_z} \to \infty$, the gradient norm diverges as $\mathcal{O}(\|\mathbf{K}_{\mathrm{eff}}\|_2^{h-1})$.

**Step 4: $\mathcal{L}_{\mathrm{KAL}}$ prevents collapse.** Under $\phi_\varepsilon$ with $\varepsilon \to 0$, for any bounded $\mathbf{K}_{\mathrm{eff}}$ we have $\mathbf{K}_{\mathrm{eff}}^h \mathbf{z}_T \approx \mathbf{0}$, so $\phi^{-1}(\mathbf{K}_{\mathrm{eff}}^h \mathbf{z}_T) \approx \phi^{-1}(\mathbf{0})$, a constant independent of $h$. The anchored KAL target (equation 30) then satisfies $f_{T+h}^{\mathrm{KAL}} \approx \tilde{f}_T$ for all $h$, which cannot match $f_{T+h}^*$ for all $h \in \mathcal{H}$ simultaneously, since target features $f_{T+h}^*$ differ across horizons by an amount proportional to the subject's motion velocity. Hence $\mathcal{L}_{\mathrm{KAL}} > c > 0$ for some constant $c$ depending on minimum inter-frame feature variation, providing a non-vanishing gradient that prevents $\varepsilon \to 0$ and keeps $\|\mathbf{B}\|_F$ bounded. $\qquad\square$

## D  Implementation Details

All models are trained on a single NVIDIA RTX 5090 GPU with the AdamW optimizer, batch size 8, and gradient clipping at 1.0 with the observation window $T = 10$ frames. Multi-horizon predictions are evaluated at $\mathcal{H} = \{1, 3, 5, 10, 15, 20\}$ frames, corresponding to 100-2000 ms at the MM-Fi sampling rate, with maximum horizon $\zeta = \max(\mathcal{H}) = 20$ frames. The model dimension is $d = 128$, Koopman latent dimension $D_z = 256$, and low-rank rank $r = 16$. The CSI encoder uses $L_c = 4$ Mamba layers; the temporal encoder uses $L_t = 2$ Mamba layers; the HPE module uses $L_p = 2$ temporal Mamba layers and $L_j = 1$ per-joint temporal block. All Mamba blocks use state dimension 16 and convolution width 4. Multi-head attention in both the HPE module and skeleton encoder uses $n_{\mathrm{head}} = 4$ heads with per-head dimension $d_h = d/n_{\mathrm{head}} = 32$. Horizon weights in $\mathcal{L}_{\mathrm{pred}}$ are set to $w_h \in \{0.3, 0.5, 0.8, 1.2, 1.5, 2.0\}$ for $h \in \{1, 3, 5, 10, 15, 20\}$, emphasising long-horizon consistency. The residual dynamics matrix $\mathbf{B}$ is initialized with entries of scale $0.5/D_z$, and its Frobenius norm $\|\mathbf{B}\|_F$ is logged as a diagnostic at every epoch.

## E   Scheduled Sampling Protocol

NEW

The skeleton-aware pose encoder receives a convex interpolation between ground-truth observed poses and stop-gradient HPE estimates:

$$\text{pose\_input}_t \; = \; \alpha \, p_t \; + \; (1 - \alpha) \, \text{sg}(\tilde{p}_t), \tag{44}$$

where $\text{sg}(\cdot)$ denotes stop-gradient and $\alpha \in [0, 1]$ is the mix ratio. The schedule spans three phases over 20 main training epochs, which follow 8 dedicated HPE pretraining epochs:

Table 4: Scheduled settings.

| Phase | Epochs | $\alpha$ | Skeleton encoder input |
|---|---|---|---|
| Warmup | 1-6 | 1.0 | Ground-truth $p_t$ |
| Decay | 7-12 | $1 - (e - 6)/6 \in (1, 0)$ | Interpolation |
| End-to-end | 13-20 | 0.0 | $\text{sg}(\tilde{p}_t)$ |

At inference $\alpha = 0$ throughout, matching the end-to-end phase. The HPE estimate $\tilde{p}_t$ is stop-gradiented before entering the skeleton encoder regardless of $\alpha$, so $\mathcal{L}_{\text{pred}}$ never backpropagates through the HPE module, only $\mathcal{L}_{\text{est}}$ trains the HPE parameters (Section 3.8).

## F   Qualitative Results

Figures 3 and 4 visualise two representative sequences from MM-Fi, showing input CSI spectrograms, ground-truth future poses (blue), and KOALA predictions (red) across horizons from $+100\,\text{ms}$ to $+2000\,\text{ms}$.

In the first sequence in Figure 3, the subject performs a relatively constrained motion involving gradual arm raising. At short horizons ($+100\,\text{ms}$ to $+500\,\text{ms}$), KOALA predictions closely align with the ground truth in both global body configuration and individual joint positions. At longer horizons ($+1000\,\text{ms}$ to $+2000\,\text{ms}$), minor deviations appear in distal joints such as the wrists and ankles, yet the overall body structure remains anatomically plausible with no structural collapse, confirming that the Koopman operator maintains coherent latent dynamics throughout the rollout.

The second sequence, as illustrated in Figure 4 involves more dynamic motion with wider limb displacement and greater inter-frame variability, presenting a substantially harder prediction target. Despite this increased articulation complexity, KOALA accurately tracks the global pose trajectory at short horizons and continues to produce structurally coherent skeletons through to $+2000\,\text{ms}$. The predicted poses preserve key kinematic constraints such as joint connectivity and limb proportions, with no catastrophic drift or physically implausible configurations at any horizon.

## G   KOALA Efficiency Metrics

NEW

To clarify the practical deployment cost of KOALA, we report its computational footprint. All measurements are on a single NVIDIA RTX 5090 with batch size 1, observation window $T = 10$, 6 prediction horizons.

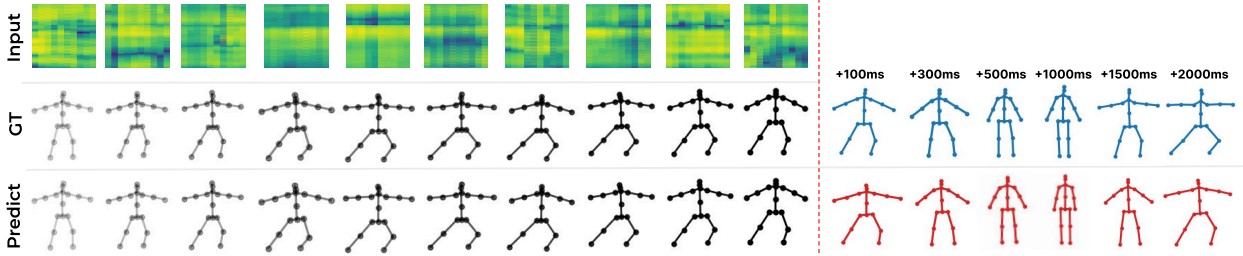

Figure 4: Additional qualitative results between predicted poses and GT for different time steps.

Table 5: KOALA efficiency metrics on MM-Fi and WiPose settings.

| Metric | MM-Fi | WiPose |
|---|---|---|
| Total Parameters | 4.8M | 4.5M |
| FLOPs | 0.083G | 0.083G |
| Inference Latency | 3.4 ms | 3.1 ms |
| GPU Memory (FP32) | 19 MB | 18 MB |
| GPU Memory (INT8) | 5 MB | 5 MB |
| FPS | ~292 | ~324 |

These figures indicate that KOALA operates well within real-time constraints for WiFi sensing (CSI sampling rate: 100-200 Hz, *i.e.*, 10-20 frames per second). With 8-bit quantization, the model size reduces to 5 MB, making edge deployment feasible.

## H  Per-Joint Error Analysis

NEW

Table 6 presents the MPJPE for individual joints on MM-Fi and WiPose. On MM-Fi, KOALA achieves the lowest errors on the pelvis and hip joints, while larger errors are observed for distal joints, particularly the elbows and wrists. These joints undergo larger motion amplitudes and are therefore more difficult to infer from WiFi CSI. A similar trend is observed on WiPose, where most torso and lower-body joints achieve relatively low errors (approximately 20-30 mm), whereas the wrists remain the most challenging body joints.

## I  Empirical Koopman Diagnostics

NEW

As discussed in Section 3.7 and the note following Theorem 1, our use of "Koopman" describes the architectural motivation for the lifting-and-linear-rollout design rather than a proven theoretical property. We measure $\epsilon_h = \|\phi(\tilde{f}_{T+h}) - \mathbf{K}_{\text{eff}}^h \phi(\tilde{f}_T)\|$ on held-out validation sequences, where $\phi(\tilde{f}_{T+h})$ is obtained by lifting the temporal-encoder feature computed from the ground-truth future pose sequence, and $\mathbf{K}_{\text{eff}}^h \phi(\tilde{f}_T)$ is the operator's $h$-step rollout from the last observed latent state. We additionally report the relative defect $\epsilon_h/\|\phi(\tilde{f}_{T+h})\|$ to disambiguate genuine divergence from a simple scale effect of the latent space. Table 7 reports both across horizons.

The relative defect grows from 0.68× at $h$=1 to 7444.4× the scale of the true target at $h$=20, indicating that the latent rollout diverges from the ground-truth trajectory by several orders

Table 6: Per-joint MPJPE (mm) of KOALA on MM-Fi and WiPose. Lower is better.

(a) MM-Fi

| Joint | MPJPE | Joint | MPJPE |
|---|---|---|---|
| Pelvis | 0.00 | R Shoulder | 51.13 |
| R Hip | 16.88 | L Shoulder | 50.49 |
| L Hip | 16.51 | R Ankle | 52.97 |
| Spine | 22.86 | L Ankle | 54.21 |
| Thorax | 41.93 | Head | 53.04 |
| R Knee | 42.59 | Neck | 53.22 |
| L Knee | 42.82 | R Elbow | 77.07 |
| | | L Elbow | 77.86 |
| | | R Wrist | 115.77 |
| | | L Wrist | 116.76 |
| | | **Mean** | **52.12** |

(b) WiPose

| Joint | MPJPE | Joint | MPJPE |
|---|---|---|---|
| R Knee | 16.42 | Nose | 22.84 |
| L Knee | 16.91 | R Ear | 23.01 |
| R Ankle | 17.63 | R Eye | 23.32 |
| R Hip | 18.58 | L Eye | 24.46 |
| L Hip | 19.30 | L Elbow | 28.53 |
| Neck | 19.56 | R Elbow | 30.04 |
| L Ankle | 20.70 | L Wrist | 35.13 |
| R Shoulder | 21.11 | R Wrist | 40.02 |
| L Shoulder | 21.62 | L Ear | 70.40 |
| | | **Mean** | **26.14** |

Table 7: Empirical Koopman invariance defect (absolute $\epsilon_h$, mean $\pm$ std, and relative to target norm) across prediction horizons.

| | 100ms | 300ms | 500ms | 1000ms | 1500ms | 2000ms |
|---|---|---|---|---|---|---|
| $\epsilon_h$ | $10.6 \pm 1.7$ | $40.3 \pm 5.4$ | $113.7 \pm 17.6$ | $1218.7 \pm 250.1$ | $12120.0 \pm 3076.8$ | $115888.3 \pm 34070.9$ |
| rel. $\epsilon_h$ | $0.68 \pm 0.11$ | $2.59 \pm 0.36$ | $7.30 \pm 1.15$ | $78.3 \pm 16.2$ | $778.7 \pm 199.8$ | $7444.4 \pm 2206.6$ |

of magnitude relative to the target itself, not merely in absolute terms. We therefore do not claim that $\mathbf{K}_{\text{eff}}$ achieves approximate Koopman invariance at long horizons.

## J  Ablations Study

### J.1  Architecture Ablations.

Table 8 reports ablation results on MM-Fi P1-S3. Removing the residual operator increases MPJPE by 3-6 mm across all horizons, confirming that the residual parametrization is necessary to escape the identity attractor and learn meaningful dynamics. Removing anchor-delta prediction produces a characteristic flat error profile (64.6-67.7 mm across horizons), consistent with the model collapsing to a near-constant pose output rather than modeling motion, exactly the degenerate solution the anchor formulation is designed to prevent. Removing dual-stream fusion keeps short-horizon performance competitive (52.7 mm at 100 ms) but degrades rapidly at longer horizons (92.8 mm at 1000 ms), indicating that CSI features provide critical dynamic information beyond what skeletal coordinates alone can supply. Removing HPE pretraining causes severe degradation (∼106 mm throughout), confirming that the two-stage schedule is essential: without a warm initialization of the pose estimator, the skeleton encoder receives uninformative inputs from the outset, preventing the Koopman operator from forming a useful latent space.

Table 8: Ablation study results of KOALA across prediction horizons. Evaluated on MPJPE (mm), lower values indicate better performance.

| Variant | 100ms | 300ms | 500ms | 1000ms | 1500ms | 2000ms |
|---|---|---|---|---|---|---|
| *w/o* Residual Operator | 55.8 | 62.4 | 67.8 | 72.1 | 69.7 | 67.4 |
| *w/o* Delta Prediction | 67.7 | 64.8 | 64.6 | 65.2 | 65.7 | 66.5 |
| *w/o* Dual Stream Fusion | 52.7 | 67.2 | 80.2 | 92.8 | 90.1 | 78.8 |
| *w/o* HPE Pretraining | 105.4 | 106.1 | 106.7 | 107.7 | 107.3 | 106.7 |
| **KOALA (Ours)** | **52.1** | **57.0** | **61.9** | **65.7** | **63.4** | **61.9** |

### J.2 Oracle Ablations.

Table 9 and Table 10 quantify the performance ceiling available to KOALA if the HPE module were replaced by ground-truth poses (oracle). On MM-Fi, the gap narrows substantially with horizon, from 26.8 mm at 100 ms to 7.1 mm at 1500 ms, indicating that the Koopman operator compensates for HPE noise more effectively as the prediction horizon grows and latent dynamics carry more of the burden relative to the initial pose estimate. On WiPose, the gap shrinks to just 5.2 mm MPJPE and 1.7 mm PA-MPJPE at 1000 ms, suggesting the HPE module achieves sufficient accuracy on this dataset that cascaded noise is not a significant bottleneck at long horizons. Across both datasets, the average $\Delta$ of 12.1 mm (MM-Fi) and 11.2 mm (WiPose) represents the remaining headroom from end-to-end pose estimation quality, motivating future work on tighter integration of pose estimation and motion prediction.

Table 9: Oracle upper bound comparison on WiPose dataset.

| Variant | Metric | 100ms | 300ms | 500ms | 1000ms | Avg |
|---|---|---|---|---|---|---|
| Oracle | MPJPE $\downarrow$ | 8.9 | 13.2 | 16.4 | 22.1 | 15.2 |
| | PA-MPJPE $\downarrow$ | 10.3 | 13.8 | 16.1 | 19.6 | 15.0 |
| KOALA | MPJPE $\downarrow$ | 26.1 | 26.0 | 26.5 | 27.3 | 26.5 |
| | PA-MPJPE $\downarrow$ | 20.8 | 20.8 | 21.0 | 21.3 | 21.0 |
| $\Delta$ (Oracle $-$ Ours) | MPJPE $\downarrow$ | 17.2 | 12.8 | 9.6 | 5.2 | 11.2 |
| | PA-MPJPE $\downarrow$ | 10.5 | 7.0 | 5.1 | 1.7 | 6.1 |

Table 10: Oracle upper bound comparison on MM-Fi dataset.

| Variant | Metric | 100ms | 300ms | 500ms | 1000ms | 1500ms | 2000ms | Avg |
|---|---|---|---|---|---|---|---|---|
| Oracle | MPJPE $\downarrow$ | 25.3 | 42.9 | 52.5 | 58.4 | 56.3 | 53.9 | 48.2 |
| | PA-MPJPE $\downarrow$ | 22.0 | 37.0 | 44.5 | 47.1 | 46.2 | 45.1 | 40.3 |
| KOALA | MPJPE $\downarrow$ | 52.1 | 57.0 | 61.9 | 65.7 | 63.4 | 61.9 | 60.3 |
| | PA-MPJPE $\downarrow$ | 43.5 | 47.0 | 50.5 | 52.8 | 51.8 | 51.1 | 49.5 |
| $\Delta$ (Oracle $-$ Ours) | MPJPE $\downarrow$ | 26.8 | 14.1 | 9.4 | 7.3 | 7.1 | 8.0 | 12.1 |
| | PA-MPJPE $\downarrow$ | 21.5 | 10.0 | 6.0 | 5.7 | 5.6 | 6.0 | 9.2 |

Table 11: Ablation study of KOALA under different hyperparameter settings. MPJPE / PA-MPJPE (mm), lower is better.

| Variant | 100ms | 300ms | 500ms | 1000ms | 1500ms | 2000ms |
|---|---|---|---|---|---|---|
| *(A) Model dimension d* | | | | | | |
| $d = 64$ | 54.16 / 45.75 | 58.74 / 48.94 | 63.32 / 52.19 | 67.55 / 54.72 | 65.28 / 53.82 | 63.59 / 52.94 |
| $d = 256$ | 52.95 / 44.12 | 56.54 / 46.48 | 60.49 / 49.30 | 64.24 / 51.82 | 62.41 / 50.84 | 61.31 / 50.40 |
| *(B) Latent dimension $D_z$* | | | | | | |
| $D_z = 64$ | 53.75 / 45.31 | 61.14 / 51.39 | 67.85 / 56.91 | 74.60 / 62.82 | 72.32 / 62.44 | 68.60 / 59.23 |
| $D_z = 128$ | 54.72 / 45.94 | 59.86 / 49.56 | 64.83 / 52.93 | 69.13 / 55.67 | 66.83 / 54.80 | 64.89 / 53.78 |
| *(C) Koopman rank r* | | | | | | |
| $r = 8$ | 55.53 / 46.36 | 59.23 / 48.90 | 63.66 / 52.10 | 67.83 / 55.08 | 65.75 / 53.96 | 64.44 / 53.46 |
| $r = 32$ | 53.23 / 44.87 | 60.79 / 50.93 | 67.59 / 56.40 | 74.08 / 61.29 | 71.23 / 60.27 | 68.05 / 58.07 |
| *(D) Encoder depth $L_c$* | | | | | | |
| $L_c = 2$ | 52.76 / 44.01 | 58.00 / 47.89 | 62.58 / 51.15 | 66.58 / 53.68 | 64.11 / 52.39 | 62.44 / 51.51 |
| $L_c = 6$ | 54.44 / 45.89 | 61.21 / 51.46 | 66.77 / 55.79 | 71.07 / 58.66 | 68.68 / 57.90 | 67.04 / 56.86 |
| **KOALA (Ours)** | **52.10 / 43.50** | **57.00 / 47.00** | **61.90 / 50.50** | **65.70 / 52.80** | **63.40 / 51.80** | **61.90 / 51.10** |

### J.3  Hyperparameter Settings.

Table 11 outlines sensitivity to key hyperparameters on Protocol P1, Setting S3. For model dimension, reducing to $d = 64$ increases MPJPE by 2-2.5 mm while $d = 256$ yields marginal gains below 1 mm at roughly double the parameter count, confirming $d = 128$ as the optimal trade-off. Latent dimension has the strongest effect: $D_z = 64$ reaches 74.6 mm at 1000 ms versus 65.7 mm for the default $D_z = 256$, confirming that a sufficiently large Koopman space is necessary to disentangle nonlinear motion modes, with $D_z = 128$ showing intermediate degradation. For Koopman rank, both $r = 8$ and $r = 32$ underperform the default $r = 16$: the former lacks capacity for CSI modulation while the latter over-parameterizes the adaptation and destabilizes long-horizon rollouts (74.1 mm at 1000 ms), identifying $r = 16$ as a sweet spot. Encoder depth is the least sensitive axis: $L_c = 2$ degrades slightly at long horizons while $L_c = 6$ increases error across all horizons, likely from overfitting. Overall, KOALA is most sensitive to latent dimension and rank, and robust to moderate variation in model dimension and encoder depth.

### J.4  Loss Ablations.

NEW

Table 12 isolates the contribution of $\mathcal{L}_{\text{est}}$ and $\mathcal{L}_{\text{pred}}$. Removing $\mathcal{L}_{\text{est}}$ produces a flat error profile, MPJPE varies by less than 0.2 mm across all horizons (132.75-132.92 mm). Without direct pose supervision, the HPE module yields uninformative estimates, so the Koopman operator has no meaningful pose-space signal to lift and instead propagates a near-static latent state.

Removing $\mathcal{L}_{\text{pred}}$ instead produces horizon-varying predictions (141.09 mm at 100 ms rising to 168.06 mm at 1000 ms), showing that $\mathcal{L}_{\text{KAL}}$ alone induces non-degenerate dynamics. However, absolute accuracy remains far below the full model (PCK@10 of ~13% vs. 65-74%), indicating that feature-space consistency alone is insufficient to calibrate predictions in pose space.

Table 12: Loss ablation study of KOALA. MPJPE / PA-MPJPE (mm), lower is better.

| Variant | 100ms | 300ms | 500ms | 1000ms | 1500ms | 2000ms |
|---|---|---|---|---|---|---|
| w/o $\mathcal{L}_{\text{est}}$ | 132.7 / 116.3 | 132.8 / 116.1 | 132.8 / 116.0 | 132.8 / 116.0 | 132.8 / 116.0 | 132.9 / 116.1 |
| w/o $\mathcal{L}_{\text{pred}}$ | 141.0 / 94.2 | 150.3 / 102.6 | 159.1 / 110.1 | 168.0 / 117.0 | 166.4 / 117.5 | 158.0 / 110.0 |
| KOALA (Ours) | **52.1 / 43.5** | **57.0 / 47.0** | **61.9 / 50.5** | **65.7 / 52.8** | **63.4 / 51.8** | **61.9 / 51.1** |

**J.5   Zero-Velocity Baseline.**                                                                    NEW

To directly address whether KOALA relatively flat error profile (Table 2) reflects genuine temporal modeling rather than slow dataset motion or heavy reliance on the anchor pose, we evaluate a zero-velocity baseline that predicts $\hat{\mathbf{p}}_{T+h} = \bar{\mathbf{p}}$ for every horizon $h$, using the same CSI-estimated anchor pose KOALA uses at inference, with no learned delta whatsoever.

Table 13 outlined the results of the zero-velocity baseline grows substantially with horizon (56.3mm at 100ms to 91.9mm at 1000ms, a spread of 22.6mm between $h=1$ and $h=20$), confirming that the underlying motion in this dataset is *not* negligible over the 2-second prediction window. KOALA own spread over the same range is only 9.8mm (52.1mm to 61.9mm), less than half that of the zero-velocity baseline, while also achieving substantially lower absolute error at every horizon (*e.g.*, 65.7mm vs. 91.9mm at 1000ms). If KOALA predictions were dominated by the anchor pose with only a negligible learned correction, its error growth would track the zero-velocity baseline growth closely, instead, the learned delta measurably suppresses the drift that copying the anchor alone already exhibits, indicating genuine temporal modeling beyond anchor reliance.

Table 13: Zero-velocity baseline ($\hat{\mathbf{p}}_{T+h} = \bar{\mathbf{p}}$, no learned delta) vs. KOALA.

| Variant | 100ms | 300ms | 500ms | 1000ms | 1500ms | 2000ms | Spread |
|---|---|---|---|---|---|---|---|
| Zero-Velocity | 56.3 | 68.9 | 80.4 | 91.9 | 89.6 | 78.9 | 22.6 |
| **KOALA (Ours)** | **52.1** | **57.0** | **61.9** | **65.7** | **63.4** | **61.9** | **9.8** |

**J.6   Predictor Ablations.**                                                                       NEW

Table 14 compares KOALA Koopman-inspired lifting and rollout against three alternatives operating on the same fused CSI-skeleton features, with classic Dynamic Mode Decomposition (DMD) (Schmid, 2010), the Linear Recurrent Unit (LRU) (Orvieto et al., 2023), and a Transformer predictor (Vaswani et al., 2017). KOALA outperforms all three at every horizon. DMD performs worst overall, degrading sharply with horizon (61.1mm at 100ms to 97.2mm at 1000ms), consistent with it fitting a single global linear operator without CSI-conditioning on a representation not optimized for it. LRU is the strongest baseline and shows a notably flat error profile across horizons (54.1-61.3mm), reflecting its built-in stability guarantee, but still trails KOALA by 2-8mm at every horizon. The Transformer degrades non-monotonically and substantially at mid-to-long horizons (up to 93.8mm), showing that nonlinear per-horizon prediction without a shared operator struggles to maintain consistency across horizons.

Table 14: Comparison across predictor formulations.

| Variant | Metric | 100 ms | 300 ms | 500 ms | 1000 ms | 1500 ms | 2000 ms |
|---|---|---|---|---|---|---|---|
| DMD (Schmid, 2010) | MPJPE ↓ | 61.1 | 72.3 | 84.4 | 97.2 | 95.6 | 86.8 |
| | PA-MPJPE ↓ | 51.6 | 62.2 | 72.7 | 83.3 | 83.9 | 75.7 |
| | PCK@20 ↑ | 88.4 | 84.4 | 80.7 | 77.2 | 77.9 | 80.0 |
| | PCK@10 ↑ | 68.9 | 65.4 | 62.2 | 59.3 | 59.9 | 61.0 |
| LRU (Orvieto et al., 2023) | MPJPE ↓ | 54.1 | 58.4 | 60.8 | 61.3 | 59.2 | 59.0 |
| | PA-MPJPE ↓ | 45.4 | 49.4 | 51.5 | 51.3 | 49.6 | 49.6 |
| | PCK@20 ↑ | 91.0 | 89.3 | 88.3 | 88.0 | 88.7 | 88.9 |
| | PCK@10 ↑ | 72.6 | 70.4 | 68.8 | 68.1 | 69.3 | 69.6 |
| Transformer (Vaswani et al., 2017) | MPJPE ↓ | 59.6 | 71.6 | 82.5 | 93.8 | 91.5 | 80.8 |
| | PA-MPJPE ↓ | 50.3 | 61.5 | 71.2 | 80.5 | 80.5 | 70.4 |
| | PCK@20 ↑ | 88.9 | 84.6 | 81.3 | 78.1 | 79.3 | 82.3 |
| | PCK@10 ↑ | 69.8 | 65.9 | 63.1 | 60.7 | 61.6 | 64.0 |
| **KOALA (Ours)** | MPJPE ↓ | **52.1** | **57.0** | **61.9** | **65.7** | **63.4** | **61.9** |
| | PA-MPJPE ↓ | **43.5** | **47.0** | **50.5** | **52.8** | **51.8** | **51.1** |
| | PCK@20 ↑ | **91.8** | **89.9** | **87.8** | **85.9** | **86.9** | **87.9** |
| | PCK@10 ↑ | **73.9** | **70.7** | **67.8** | **65.1** | **66.5** | **67.7** |

