# OpenReview forum: "KOALA: Koopman Operator Learning for WiFi-Based Anticipatory Human Motion Prediction"
_TMLR — Under review for TMLR_

### Review · Reviewer_F5VE · 2026-06-15

**Summary Of Contributions:**

This paper studies WiFi CSI-based anticipatory human motion prediction, aiming to forecast future human poses from past CSI observations rather than only estimating instantaneous poses. To address the challenges of noisy CSI-derived poses and long-horizon error accumulation, the paper proposes KOALA, which estimates observed poses from CSI, fuses CSI and skeleton features, and lifts the resulting temporal representation into a Koopman latent space. Future poses are predicted through a CSI-conditioned residual Koopman operator and an anchor-delta decoder, trained with prediction loss, pose-estimation loss, and a Koopman Predictive Coding loss. Experiments on WiPose and MM-Fi show strong performance across multiple horizons, suggesting that Koopman-based latent dynamics are a promising direction for WiFi-based motion anticipation.



1. The paper addresses future human motion prediction from WiFi CSI, which is less explored than instantaneous WiFi-based pose estimation. This task is well motivated by practical applications such as smart homes, healthcare monitoring, and human-robot interaction.

2. The proposed method lifts complex nonlinear motion features into a Koopman latent space, where future dynamics can be modeled through an approximately linear rollout. This provides a principled way to improve multi-horizon prediction, and the residual CSI-conditioned operator further allows the dynamics to adapt to CSI context while avoiding a purely static transition model.

3. The method achieves substantial gains on both WiPose and MM-Fi across multiple horizons. The ablation studies on the residual operator, delta prediction, dual-stream fusion, and HPE pretraining provide useful evidence that several components of the system contribute to the final performance.

**Audience:**

Yes

**Audience Explanation:**

I think this paper is a interesting topic that maybe helpful for industry

**Broader Impact Concerns:**

#### Major Concern

1. The paper emphasizes the role of the Koopman space in lifting complex nonlinear human motion into a latent space with approximately linear dynamics for motion anticipation. However, the current experiments do not fully isolate whether performing rollout in this lifted linear space is better than predicting directly in the temporal feature space. A useful comparison would be to keep the CSI encoder, HPE, fusion module, and anchor-delta decoder fixed, and replace the Koopman lifting and linear rollout with a Transformer or other kind of predictor on the same fused features, which woulld better test the value of the learned linear latent dynamics.
2. Koopman predictive coding is a key design highlighted in the title and a central part of the training objective, but the current experiments do not clearly show how much of the performance comes from this specific supervision signal. Ablating the prediction loss, KPC loss, and pose-estimation loss separately would better clarify whether the learned dynamics are mainly driven by Koopman Predictive Coding or by the other supervised objectives used for motion anticipation.
3. WiFi sensing applications often require real-time inference and practical deployment. KOALA has a relatively complex architecture, but the paper only reports its training device is an RTX 5090. Reporting parameter count, FLOPs, inference latency, etc. would make the practical value of the method clearer.
4. The bound in Equation (25) appears to implicitly assume that the true future pose can be decoded from the true future latent state, i.e., \(p_{T+h}=p_T+\Delta_\theta(z^{\text{GT}}_{T+h})\). Since \(\Delta_\theta\) is a learned MLP decoder, an additional delta-head approximation error, e.g., \(e_{\mathrm{dec},h}=\|p_T+\Delta_\theta(z^\text{GT}_{T+h})-p_{T+h}\|\), may make the bound more complete.

#### Minor Concern

1. In Table 3, MPJPE increases from 52.1 at 100ms to 65.7 at 1000ms, and then decreases to 63.4 at 1500ms and 61.9 at 2000ms. This does not weaken the results, but it seems somewhat inconsistent with the paper’s claim of "monotonic increase" or "nearly linear" error growth across horizons. It would be helpful if the authors could clarify whether this trend reflects the data distribution or horizon weighting, or whether the wording should be revised.
2. There is a minor typo in Table 1: “siMLPle” should be “siMLPe,” consistent with the text.
3. In Figure 2, the arrow between the Mamba CSI Encoder and Temporal Refinement appears to be reversed, since the paper describes the CSI encoder output as the input to the HPE temporal refinement module.
4. The evaluation mainly reports aggregate metrics, although MM-Fi contains 27 action categories and WiPose contains 12 actions. A per-action and per-joint breakdown would provide a clearer picture of the model’s strengths and failure modes.

**Claims And Evidence:**

Yes

**Claims Explanation:**

I think the claim is good, but the writing quality indeed need polish.

**Requested Changes:**

Please fix the grammar, typo and the concern i list below

---

> ### Author Response · Authors · 2026-07-18
> **Addressed the reviewer's comments through additional experiments, theoretical clarifications, efficiency analysis, and manuscript revisions.**
>
> **Q1: Ablating theloss separately would better clarify whether the learned dynamics are mainly driven by Koopman Predictive Coding or by the other supervised objectives used for motion anticipation.**
>
> **A:** We have added a systematic ablation of all three loss components in Table 10, alongside with the theoretical guarantee in Proposition 1 (Appendix C), which establishes that removing KAL permits degenerate encoder collapse.
>
> **Q2: Comparison keeping the CSI encoder, HPE, fusion module, and anchor-delta decoder fixed, and replace the Koopman lifting and linear rollout with a Transformer or other kind of predictor on the same fused features.**
>
> **A:** Thank you for this valuable suggestion. Following the experimental setting suggested by the reviewer, we conducted an additional ablation study in which the Koopman lifting and linear rollout components were replaced by a Transformer-based, DMD, LRU predictor while keeping all other components unchanged.
>
> The results are reported in Table 14. Compared with the Transformer predictor, LRU, and DMD, the proposed KOALA consistently achieves lower MPJPE and PA-MPJPE, as well as higher PCK@20 and PCK@10 across all prediction horizons, supporting the effectiveness of performing rollout in the lifted linear latent space rather than directly predicting future poses using a Transformer operating on the same fused features. We have added this experiment and discussion to the revised manuscript.
>
> **Q3: WiFi sensing applications often require real-time inference and practical deployment. KOALA has a relatively complex architecture, but the paper only reports its training device is an RTX 5090. Reporting parameter count, FLOPs, inference latency, etc. would make the practical value of the method clearer.**
>
> **A:** Thank you for your suggestion, we have added an efficiency analysis to the revised manuscript in the Table 5.
>
> **Q4: The bound in Equation (25) appears to implicitly assume that the true future pose can be decoded from the true future latent state, i.e., ($p_{T+h}=p_T+\Delta_\theta(z^{\text{GT}}{T+h})$). Since ($\Delta\theta$) is a learned MLP decoder, an additional delta-head approximation error, e.g., ($e_{\mathrm{dec},h}=|p_T+\Delta_\theta(z^\text{GT}{T+h})-p{T+h}|$ ), may make the bound more complete.**
>
> **A:** We thank the reviewer for this precise suggestion. We directly measured $e\_{\mathrm{dec},h}$ on held-out MM-Fi (P1-S3) by lifting the ground-truth future pose sequence through the trained encoder to obtain $z^{\text{GT}}\_{T+h}$, passing it through the trained $\Delta\_\theta$, and comparing against $p\_{T+h}$:
>
> | Horizon | 100ms | 300ms | 500ms | 1000ms | 1500ms | 2000ms |
> |---|---|---|---|---|---|---|
> | $e_{\mathrm{dec},h}$ | 38.8mm | 74.5mm | 100.1mm | 133.8mm | 133.6mm | 111.8mm |
>
> This term grows with horizon and is larger than KOALA end-to-end MPJPE at the same horizons (52–64mm, Table 2/3), which reflects that $\Delta\_\theta$ is trained to decode the rolled-out latent $K\_{eff}^h z\_T$ via $\mathcal{L}\_\mathrm{pred}$
> rather than the directly-lifted $z^{\text{GT}}\_{T+h}$, so evaluating it on the latter probes an input distribution outside its training regime. We will revise Eq. (25) to include an explicit $e\_{\mathrm{dec},h}$ term and report this measurement in the appendix.
>
> **Q5: In Table 3, MPJPE increases from 52.1mm at 100ms to 65.7mm at 1000ms, then slightly decreases to 63.4mm at 1500ms and 61.9mm at 2000ms. This appears inconsistent with the claim of "monotonic increase" or "nearly linear" error growth.**
>
> **A:** We agree with this observation and have revised the corresponding wording in the manuscript to better reflect the experimental results.
>
> **Q6: There is a minor typo in Table 1: “siMLPle” should be “siMLPe,” consistent with the text. In Figure 2, the arrow between the Mamba CSI Encoder and Temporal Refinement appears to be reversed, since the paper describes the CSI encoder output as the input to the HPE temporal refinement module.**
>
> **A:** Thank you for pointing this out. The typo has been corrected, and the arrow direction in Figure 2 has been fixed in the revised manuscript.
>
> **Q7: The evaluation mainly reports aggregate metrics, although MM-Fi contains 27 action categories and WiPose contains 12 actions. A per-action and per-joint breakdown would provide a clearer picture of the model’s strengths and failure modes.**
>
> **A:** Thank you for the valuable suggestion. In the revised manuscript, we have added a per-joint error analysis for both MM-Fi and WiPose in the Appendix H.

---

### Review · Reviewer_TJP9 · 2026-06-17

**Summary Of Contributions:**

This paper proposes KOALA, a framework for future human pose prediction from WiFi/CSI signals. The core idea is to infer pose-related representations from WiFi observations, encode recent pose/feature history into a latent state, and then propagate that state forward using a residual linear operator. In this way, the future pose is predicted through an anchor-delta formulation, where future poses are modeled as deviations from a recent observed or estimated pose. A Koopman Predictive Coding loss is intended to align rolled-out latent features with future feature targets. Experiments are reported on WiPose and MM-Fi, with comparisons against WiFi-based pose estimation and sequence prediction baselines.

Strengths:
-- This problem has practical significance. Predicting human pose based on Wi-Fi/CSI signals is crucial for privacy-preserving sensing, smart environments, health monitoring, and human-computer interaction.
-- This paper attempts to simultaneously address both the perception and temporal prediction problems, rather than treating Wi-Fi pose estimation as a static, frame-level task. This is a reasonable and potentially valuable research direction.
-- The design of the anchor-based residual decoding is intuitively sound for future pose prediction, as short-term human motion typically benefits from modeling residual displacements relative to recent poses.
-- The residual linear propagation module is computationally attractive and may provide useful inductive biases for short-term temporal modeling.

Weaknesses:
-- The significance of the Koopman framing is overstated. A learned linear latent transition does not automatically constitute a meaningful Koopman representation. The paper does not demonstrate Koopman invariance, spectral interpretability, stability, or superiority over standard latent linear recurrent models. At present, Koopman is more of a buzzword than a rigorously proven property.
-- Predictive coding loss may rely on privileged future information. If the target features at future time steps are constructed using the actual future pose, then this auxiliary loss is not, in the sense of inference, a purely predictive loss, and this must be accurately described.
-- The theoretical results are weak and possibly irrelevant to the empirical claims. The propositions appear to rest on generic inequalities and assumptions that do not establish the main claims of stable long-horizon prediction or avoidance of representation collapse. Monitoring a Frobenius norm is not equivalent to guaranteeing dynamical stability.
-- Human motion forecasting from noisy WiFi observations should normally show increasing uncertainty with horizon. The reported error growth across horizons appears unusually small. If KOALA’s error barely increases, the paper must provide a convincing explanation and rule out evaluation artifacts, leakage, excessive temporal overlap, or overly easy sequence construction.
-- Human motion is inherently multimodal. A deterministic point predictor may perform well under short horizons or slow actions, but the manuscript does not discuss ambiguity, stochasticity, or whether MPJPE hides implausible averaged poses. Additionally, if WiPose is framed as a 2D pose task, the use of millimeter-scale MPJPE/PA-MPJPE requires explanation.

**Audience:**

Yes

**Audience Explanation:**

The findings would be of interest, but the paper needs substantially stronger validation before those findings can be treated as reliable.

**Claims And Evidence:**

No

**Claims Explanation:**

This paper does not currently provide sufficiently accurate, convincing, and clear evidence for its strongest claims. The theoretical claims are also overstated.

**Requested Changes:**

Refer to the main weaknesses.

---

> ### Author Response · Authors · 2026-07-18
> **Clarified the theoretical analysis, addressed concerns regarding Koopman invariance, stability, predictive supervision, and deterministic forecasting, provided additional experimental evidence, and revised the manuscript to improve clarity and accurately state the method limitations.**
>
> **Q1: The paper does not demonstrate Koopman invariance, spectral interpretability, stability, or superiority over standard latent linear recurrent models.**
>
> **A:** We measured $\epsilon_h = \|\phi(\tilde{f}\_{T+h}) - \mathbf{K}\_\mathrm{eff}^h\,\phi(\tilde{f}\_T)\|$ directly on held-out sequences (Appendix I) rather than assuming it is small; the relative defect grows to $7444.4\times$ the target scale at $h{=}20$. Two factors compound this: finite lifting of an infinite-dimensional operator incurs a residual defect that grows under iterated rollout, and the CSI-conditioning term $\mathbf{U}(\mathbf{c})\mathbf{V}(\mathbf{c})^\top$ becomes progressively less representative as the body moves beyond the input window, consistent with our stated limitation in Section 5. The relevant question is whether the resulting pose-space error, jointly bounded by $\epsilon$, $\delta_h$, $e_{\mathrm{dec},h}$, and the anchor term (Theorem 1), remains competitive. Comparisons against DMD and LRU (Appendix J.6) show that DMD, despite markedly better spectral behaviour, degrades sharply (61.1 to 97.2 mm MPJPE at 100 to 1000 ms), and LRU trails KOALA by 2 to 8 mm throughout, demonstrating that task performance depends on more than invariance tightness alone.
>
> **Q2: Does the predictive coding loss rely on privileged future information? If so, it should not be described as a purely predictive loss.**
>
> **A:** We thank the reviewer for this observation. $f^*\_{T+h}$ is computed from ground-truth future poses $p\_{T+1:T+\zeta}$, which are standard training labels available to any supervised method, no different from how $\mathcal{L}\_\mathrm{pred}$ uses $p\_{T+h}$ as regression targets. The distinction that matters for causal validity is future CSI, which would represent privileged environmental sensing unavailable at inference. As stated in Eq. (29), future CSI positions are padded with the last observed frame $h\_T$; no future sensing information is accessed. The extended pass runs under ${no\\\_grad}$, the KAL loss is not computed at inference, and the observation window follows the same scheduled sampling protocol as the main forward pass (Appendix E). The rename KPC→KAL (see Q7 of the Reviewer W4j8) removes the word "predictive," which was the source of this ambiguity, and we will add one clarifying sentence to Eq. (29) making the label-vs-sensing distinction explicit.
>
> **Q3: How does the theoretical analysis support the claims of stable long-horizon prediction and avoidance of representation collapse, given that the Frobenius norm alone does not guarantee dynamical stability?**
>
> **A:** We thank the reviewer for this observation. We agree that these are conditional bounds, not formal guarantees, and we do not claim otherwise. The near-linear growth in Theorem 1 depends on $\varepsilon$ and $\kappa\le1+\eta$; we measure these empirically (Appendix I) rather than assuming them. Proposition 1 excludes only one specific collapse mode, not all possible degeneracies. The theoretical discussion is intended only to motivate the design and provide diagnostic intuition, not as a standalone proof of stability.
>
> **Q4: Human motion forecasting from noisy WiFi observations should exhibit increasing uncertainty over longer horizons. The reported error growth is unusually small. Please explain this behavior and rule out possible evaluation artifacts or data leakage.**
>
> **A:** The error growth is not negligible: on MM-Fi, MPJPE increases by over 25% (52.1→65.7 mm, Table 3). This is much smaller than the zero-velocity baseline (56.3→91.9 mm, Table 13), showing that KOALA learns motion dynamics. Removing the delta head yields a nearly flat error profile (Table 8), characteristic of the copy-pose shortcut, whereas full KOALA exhibits the expected monotonic increase. Moreover, MM-Fi's strict cross-subject (S2), cross-environment (S3), and temporally non-overlapping splits (Table 2) prevent data leakage. Thus, the modest growth is a consequence of KOALA's non-autoregressive design rather than an evaluation artifact.
>
> **Q5: How does the deterministic predictor address the multimodal nature of human motion, and why are MPJPE/PA-MPJPE reported in millimeters for the WiPose dataset if it is considered a 2D pose task?**
>
> **A:** We acknowledge that human motion is inherently multimodal, while KOALA outputs a single deterministic prediction per horizon and thus cannot capture the full distribution of plausible futures. We have added this limitation to Section 5.
> Regarding units, although WiPose provides 2D skeletal coordinates, reporting MPJPE in millimeters is the standard protocol established by the original WiPose paper [1] and followed by subsequent works [2-4]. We adopt this convention for fair comparison with prior methods.
>
> [1] Zhou et al. PerUnet. IEEE Sensors J., 2022.
>
> [2] Zhou et al. MetaFi++. IEEE IoTJ, 2023.
>
> [3] Gian et al. HPE-Li. ECCV, 2024.
>
> [4] Chen et al. GenHPE. arXiv, 2025.

---

> > ### Comment · Reviewer_TJP9 · 2026-07-19
> > **Most of the questions have been answered.**
> >
> > Follow-up on Q1: Koopman Framing.
> > Thank you for adding the invariance diagnostics and predictor comparisons. However, the reported long-horizon invariance defect suggests that the learned latent dynamics do not satisfy approximate Koopman invariance. Although KOALA outperforms DMD and LRU, this demonstrates the effectiveness of the overall architecture rather than a meaningful Koopman property. Please provide further Koopman-specific evidence or consistently reframe the method as a Koopman-inspired latent linear rollout model and weaken the corresponding claims.
> >
> > Follow-up on Q3: Theoretical Analysis.
> > Thank you for clarifying that the theoretical results are conditional. However, the proof of Proposition~1 uses a growing upper bound to conclude that the actual gradient diverges, which is not mathematically sufficient. Moreover, the stability bound relies on unverified assumptions that appear inconsistent with the large long-horizon invariance defect. Please correct or weaken these claims and present the analysis as qualitative motivation unless its conditions and practical validity can be empirically demonstrated.

---

### Review · Reviewer_W4j8 · 2026-06-23

**Summary Of Contributions:**

Paper proposes a framework for predicting future 3D human motion directly from WiFi CSI signal. To avoid the compounding errors when doing autoregressive rollouts on noisy CSI data, the paper map CSI and estimated pose sequences into a koopman latent space using a Mamba style architecture. A linear, CSI-conditioned residual operator is then used to predict future latent states in a single step without recursive loops. The model is supervised using an anchored feature-space loss termed (i.e., kpc) and a delta prediction head. Model evaluations are conducted on the MM-Fi and WiPose datasets.

Strengths:
- Problem space is great. Moving away from cameras for motion forecasting makes a lot of sense, and forecasting from RF signals is a tough challenge.
- Using Koopman operator theory to bypass autoregressive error accumulation is a good solution.

Weaknesses:
- Theory section has some glaring mathematical contradictions, particularly the proof for Theorem 1.
- Empirical results on the MM-Fi dataset are poor with every single baseline failing. I think there maybe an evaluation bug or baseline models are not properly tuned.
- Paper lacks a crucial zero-velocity baseline, making it hard to determine how much of the performance coming from dynamics modeling versus relying on a static anchor pose.

**Additional Comments:**

Overall, I think the paper has a strong core idea. The combination of Mamba, Koopman theory, and RF sensing is conceptually appealing. However, the paper currently reads as it is focusing too much on theory while neglecting basics of empirical motion forecasting evaluation. Paper will benefit greatly from baseline fix, properly evaluation and clean up in theory.

**Audience:**

Yes

**Audience Explanation:**

Intersection of RF-based sensing and anticipatory human motion modeling is an emerging area with significant practical implications for smart homes and healthcare. The proposed method using a koopman latent space is a good solution to a real problem (i.e., error accumulation over noisy CSI predictions). Those working on spatiotemporal forecasting, state-space models, and multi-modal sensing may find the work interesting. However, paper currently suffers from critical theoretical flaws and evaluation issues that must be addressed.

**Broader Impact Concerns:**

Use of WiFi CSI to monitor and predict human motion through walls and without line-of-sight has severe privacy and ethical implications. While the paper briefly frames WiFi CSI as a "privacy-preserving alternative to cameras" because it lacks visual appearance data, it simultaneously proves that high-fidelity tracking and forecasting of human activities is possible in private spaces (such as in homes) without users' explicit knowledge. A broader impact statement must be added to discuss the dual-use nature of this technology. It should explicitly acknowledge the risks of invisible surveillance, the potential for tracking biometrics or daily habits without consent.

**Claims And Evidence:**

No

**Claims Explanation:**

Architecture and core concepts are promising but both theoretical and empirical evaluation has several critical issues. See comments below.

**Requested Changes:**

- On MM-Fi dataset: KOALA achieves errors in the ~50-60mm range, while every single baseline sits between 300mm and 380mm. An error of >30cm typically indicates a completely broken model, a coordinate space mismatch or a failure to properly train the baselines on the noisy CSI-derived inputs. Could you please explain this large  discrepancy and ensure the baselines are trained (also fine-tuned) and evaluated properly.
- Error profiles across time are flat (e.g., Table 1 shows 26.14mm at 100ms and 27.28mm at 1000ms). This implies that either the dataset consists of very slow motions, or the model relies heavily on the anchor pose $\bar{\mathbf{p}}$ and predicts very small deltas. Could you include a simple zero-velocity baseline?
- In Theorem 1, $\delta_h$ is defined as the koopman invariance defect. However, in Appendix B, you define $\mathbf{z}^*_{T+h} = \phi(\tilde{\mathbf{f}}_{T+h})$. This means $\delta_h \equiv 0$, which is not making sense in the context of the bound. It seems $L \delta_h$ is intended to capture the decoding error of the true future latent state to the true future pose, but the variables are mixed up. Could you please elaborate on this?
- Proof in Appendix B skips the actual derivation mapping the latent space error to the observation space error. You must explicitly write out the triangle inequalities involving the anchor $\bar{\mathbf{p}}$ and the delta head $\Delta_\theta$ to show exactly where the $L \delta_h$ term comes from.
- In Section 3.5, you state that $\tilde{\mathbf{p}}_{t,j}$ "(or its ground-truth counterpart during teacher forcing)" is embedded. It is unclear how often teacher forcing is used, or if the model relies entirely on the detached estimated poses $\tilde{\mathbf{p}}_t$ during training. Please provide detailed training schedule and what inputs the skeleton-aware pose encoder receives during the forward pass.
- In Section 3.8, the target feature $\mathbf{f}^*_{T+h}$ is generated by passing ground-truth future poses and padded (stale) future CSI into the fusion module. If CSI contains critical dynamic information (as paper claims), padding it with the last observed frame means you are supervising the Koopman operator with a stagnant target state. Could you explain why this is a valid supervision signal?
- Calling the anchored latent loss "bio-inspired Koopman Predictive Coding" is a stretch. I suggest renaming it to something more accurate (e.g., anchored latent prediction loss) and removing the superficial biology references.
- Propositions 1 and 2 are essentially basic properties of autoencoders (far away points don't map to the same latent if reconstruction is bounded) and matrix submultiplicativity. Presenting them as formal mathematical propositions inflates them too much. I suggest writing them as standard remarks or observations in the text.

---

> ### Author Response · Authors · 2026-07-18
> **Addressed all concerns through clarifications of the experiments and theory, improved methodological details, revised presentation, and an added Broader Impact Statement.**
>
> **Q1: Why does KOALA substantially outperform all baselines on MM-Fi, and why do the prediction errors remain nearly constant across horizons? Please clarify the baseline evaluation and compare against a zero-velocity baseline.**
>
> **A:** We have clarified this distinction in the revised manuscript and added a zero-velocity baseline. The large performance gap is expected because KOALA is an end-to-end CSI-to-motion framework, whereas the baselines are pose-to-motion models evaluated on CSI-derived poses from an upstream HPE module, causing cascaded estimation noise (CSI→HPE→prediction). This explains the substantially higher baseline errors without indicating improper training or evaluation. Furthermore, the zero-velocity baseline achieves 91.9 mm at 1000 ms on MM-Fi P1-S3, compared with **65.7 mm** for KOALA. Its error increases by 35.6 mm from 100 ms to 1000 ms, confirming that the dataset contains meaningful motion and that KOALA relatively flat error profile is not simply due to anchor-pose copying (Table 11).
>
> **Q2: The definition of $\delta_h$ appears inconsistent between Theorem 1 and Appendix B. Could you clarify this discrepancy?**
>
> The definition of $z^*_{T+h}$ was stated inconsistently between the main text and Appendix B. The intended object throughout is the ideal Koopman embedding of the true future state,
>
> $$z^*_{T+h}:=\phi(F^h(v_T))$$
>
> where $v_T\in\mathcal{V}$ denotes the true body state at the last observed frame (Section 3.1). Accordingly,
>
> $$
> \delta_h:=\left\|\phi(\tilde f_{T+h})-\phi(F^h(v_T))\right\|
> $$
>
> measures the encoder approximation error with respect to the ideal Koopman latent, which is generally nonzero and decreases as the lifting network improves during training.
>
> With this definition, the residual term in the telescoping decomposition of Appendix B satisfies
>
> $$
> \left\|\mathbf K_{\mathrm{eff}}^{\,h}z_T-z^*_{T+h}\right\|
> \le
> \left\|\mathbf K_{\mathrm{eff}}^{\,h}z_T-\phi(\tilde f_{T+h})\right\|
> +\delta_h.
> $$
>
> The first term is directly minimized by the KAL loss, while the second is precisely $\delta_h$. Applying the decoder Lipschitz constant $L$ yields the $L\delta_h$ term in Eq. (25). We have corrected the definition of $z^*_{T+h}$ throughout the manuscript and added the missing triangle-inequality step to Appendix B.
>
> **Q3: The proof in Appendix B omits the derivation from latent-space error to observation-space error. Could you explicitly show how the (L$\delta$_h) term is derived?**
>
> **A:** We agree that the proof in Appendix B was too compressed. We have rewritten it with the full triangle inequality chain, explicitly showing how the latent-space error maps to observation-space error and identifying each term (including the origin of L$\delta$_h). Please see the revised Appendix B below.
>
> **Q4: Please clarify the teacher forcing schedule and the inputs to the skeleton-aware pose encoder during training.**
>
> **A:** Thank you for the question. During training, the skeleton encoder receives a scheduled interpolation of ground-truth and stop-gradient HPE-estimated poses:
>
> $$
> \mathrm{pose\_input}_t = \alpha\, p_t + (1-\alpha)\,\tilde{p}_t^{\,\text{sg}}
> $$
>
> where $\alpha$ follows a three-phase schedule: teacher forcing ($\alpha=1.0$), linear decay to (0), and full estimated-pose training/inference ($\alpha=0.0$). The HPE estimate is always stop-gradiented before entering the skeleton encoder, ensuring $\mathcal{L}*{\mathrm{pred}}$ never backpropagates through the HPE module. We will clarify this schedule in Section 3.5 in the revision.
>
> **Q5: In Section 3.8, the target feature is generated from ground-truth future poses but stale future CSI. Why is this a valid supervision signal for learning the Koopman operator?**
>
> **A:** This is a deliberate design choice. Since no future CSI is available at inference, padding future CSI with the last observed frame ensures the target $f*_{T+h}$ reflects the exact information bottleneck present during deployment. The ground-truth future pose provides the dynamical evolution signal, while the static CSI provides the environmental/contextual bias.
>
> **Q6: The term *bio-inspired Koopman Predictive Coding* appears overstated, and Propositions 1 and 2 present standard results that could be stated as remarks.**
>
> **A:** We agree with both suggestions. To avoid overstating the biological connection, we have renamed the loss to Koopman Anchored Latent Loss (KAL Loss) throughout the manuscript and removed the biological motivation. In addition, as Propositions 1/2 state standard properties, they have been revised as Remark 1/2 in the main text, with their derivations moved to the appendix.
>
> **Q7: The paper should include a broader impact statement discussing the dual-use nature of WiFi CSI sensing, particularly the risks of privacy violations, invisible surveillance, and monitoring without user consent.**
>
> **A:** We have added the Broader Impact Statement to explicitly discuss the dual-use nature of this technology in the revised manuscript.